# $N^6$-methyladenosine of Spi2a attenuates inflammation and sepsis-associated myocardial dysfunction in mice

Xiangyu Wang[1,2,3,10], Yan Ding[4,5,10], Ran Li[1,2,3,10], Rujun Zhang[6,7,10], Xuejun Ge[1,8], Ruifang Gao[1], Miao Wang[6,7], Yubing Huang[6,7], Fang Zhang [1,3], Bin Zhao[1], Wang Liao [6,7] ✉ & Jie Du [1,3,9] ✉

Bacteria-triggered sepsis is characterized by systemic, uncontrolled inflammation in affected individuals. Controlling the excessive production of pro-inflammatory cytokines and subsequent organ dysfunction in sepsis remains challenging. Here, we demonstrate that Spi2a upregulation in lipopolysaccharide (LPS)-stimulated bone marrow-derived macrophages reduces the production of pro-inflammatory cytokines and myocardial impairment. In addition, exposure to LPS upregulates the lysine acetyltransferase, KAT2B, to promote METTL14 protein stability through acetylation at K398, leading to the increased m6A methylation of Spi2a in macrophages. m6A-methylated Spi2a directly binds to IKKβ to impair IKK complex formation and inactivate the NF-κB pathway. The loss of m6A methylation in macrophages aggravates cytokine production and myocardial damage in mice under septic conditions, whereas forced expression of Spi2a reverses this phenotype. In septic patients, the mRNA expression levels of the human orthologue SERPINA3 negatively correlates with those of the cytokines, TNF, IL-6, IL-1β and IFNγ. Altogether, these findings suggest that m6A methylation of Spi2a negatively regulates macrophage activation in the context of sepsis.

Sepsis, which is defined as the systemic inflammatory response syndrome (SIRS) occurring upon infection, is commonly considered a disease triggered by the uncontrolled immune reaction in the affected individuals[1–4]. Sepsis, the main cause of patients' death in intensive care units (ICU) across the world, accounts for 6% of adult hospitalizations[5]. Every year, the rate of sepsis incidence and sepsis-related mortality seems to be increasing by 1.5% on average[6]. This disease is featured severe systemic inflammatory response and organ failure due to the dysregulated immune responses to bacterial or viral infection[7]. Although the rapid cytokine storm initiated by infection is helpful for the host to eliminate the infectious agent, the excessive inflammatory response is able to result in a detrimental systemic inflammatory response syndrome and subsequently cause organ dysfunction[7]. Pro-inflammatory cytokines (TNF, IL-6, IL-1β, and IFNγ) have important functions in sepsis development[8]. Macrophages exert critical effects on the host immune defense against infection[9], but can also initiate a

[1]Shanxi Province Key Laboratory of Oral Diseases Prevention and New Materials, Shanxi Medical University School and Hospital of Stomatology, Taiyuan, Shanxi, China. [2]Department of Child Dental and Preventive Dentistry, Shanxi Medical University School and Hospital of Stomatology, Taiyuan, Shanxi, China. [3]Department of Oral Medicine, Shanxi Medical University School and Hospital of Stomatology, Taiyuan, Shanxi, China. [4]Department of Dermatology, Hainan Provincial Hospital of Skin Disease, Haikou, Hainan, China. [5]Department of Dermatology, Skin Disease Hospital of Hainan Medical University, Haikou, Hainan, China. [6]Department of Cardiology, Hainan General Hospital and Hainan Affiliated Hospital of Hainan Medical University, Haikou, China. [7]Hainan Clinical Research Center for Cardiology, Haikou, China. [8]Department of Endodontics, Shanxi Medical University School and Hospital of Stomatology, Taiyuan, Shanxi, China. [9]Institute of Biomedical Research, Shanxi Medical University, Taiyuan, Shanxi, China. [10]These authors contributed equally: Xiangyu Wang, Yan Ding, Ran Li, Rujun Zhang. ✉e-mail: crain_lw@163.com; dj1243@hotmail.com

cytokine storm amid bacterial infection[9]. In humans, bacteria-activated macrophages produce plenty of pro-inflammatory cytokines and chemokines which initiate an overwhelming inflammatory reaction[9]. Thus, controlling the overwhelming systemic inflammation and reducing excessive cytokines are important steps for sepsis amelioration.

Recently, a growing number of studies have reported that chemical modifications on RNA are able to regulate gene expression. Among the numerous identified chemical marks on RNA nucleotides, $N^6$-methyladenosine (m$^6$A) methylation is the most prevalent in eukaryotic mRNAs[10,11]. m$^6$A modification is reversible and plays a regulatory role in over 7000 mRNAs in transcriptomes of mammalian cells[12,13]. This modification exists on adenosines with consensus motif RRm$^6$ACH [(G/A/U)(G > A)m$^6$AC(U > A > C)] mostly near the stop codons[14]. The m$^6$A methyltransferase complex including methyltransferase-like 3 (METTL3), methyltransferase-like 14 (METTL14), and Wilms tumor 1-associating protein (WTAP) deposits m$^6$A modification on RNAs[15,16]. Demethyltransferase of m$^6$A, such as fat mass and obesity-associated protein (FTO) and AlkB Homolog 5 (ALKBH5), removes methyl chemical groups from adenosines reversibly[17]. The identified m$^6$A modification readers involving the heterogeneous nuclear ribonucleoprotein (HNRNP) protein family (HNRNPA2B1 and HNRNPC), YT521-B homology (YTH) domain-containing protein family (YTHDF1/2/3 and YTHDC1/2) and the insulin-like growth factor 2 mRNA-binding proteins (IGF2BP1/2/3) recognize m$^6$A and regulate its functions[18–21]. Cytoplasmic YTHDF2 facilitates the target transcripts' degradation, while YTHDF1 and YTHDF3 are demonstrated to enhance the translation rate of target transcripts[17]. YTHDC1 has been pointed to regulate mRNA splicing and YTHDC2 modulates mRNA stability and translation[17]. In addition, IGF2BPs are able to stabilize target mRNAs[17]. Increasing evidence has indicated that RNA m$^6$A modification tunes RNA fate and functions and is prominent for most bioprocesses such as circadian rhythm, tissue development, DNA damage response, tumorigenesis, and sex determination[22].

NF-κB is a group of transcription factor-associated proteins including p105/p50, p100/p52, p65, c-Rel, and RelB[23,24]. The inhibitor of κB (IκB) acts as a negative regulator to suppress NF-κB activation by binding to p65 and influencing the movement of p65/p50 heterodimer[23]. IκB protein stability is regulated by the IKK complex consisting of IKKα, β, and γ subunits[23].

Lysine acetylation, a post-translational modification, has emerged as a prominent metabolic regulatory mechanism[25]. Lysine acetylation is catalyzed by the protein acetyltransferase (KAT) which transfers the acetyl group to lysine residues[25–27]. The acetyl group on lysine is removed by deacetylase enzymes such as sirtuins (SIRTs) and histone deacetylases (HDACs)[28,29].

Serpins, which are known as a protein superfamily, regulate serine and cysteine protease functions in various processes such as fibrinolysis, extracellular matrix degradation, coagulation, complement activation, and apoptosis[30]. Serine protease inhibitor 2 A (Spi2a), also known as Serpin Family A Member 3 g (*Serpina3g*), is encoded by the *Serpina3g* gene on mouse chromosome 12 and resides in both cytosol and nucleus[31,32]. Spi2a is identified as an inhibitor of serine proteases and cysteine cathepsins[33]. Although there is no mouse *Serpina3g* homolog in humans, human cells express the ova-serpins SERPINB3 and SERPINB4 to inhibit serine proteases and cysteine cathepsins[31]. In addition to the role of serine protease inhibitor, Spi2a's functions in immune cells attract more and more attention recently. Studies have suggested that Spi2a contributes to memory T cell development, controls the viability of Th2 cells, and mediates B-cell and granulocyte formation[34–37]. *SERPINA3*, the human orthologue of Spi2a, is also identified to be an inflammation-related gene and closely associated with immune responses[38,39]. However, the function of Spi2a in macrophages in the context of sepsis remains elusive. Here we report Spi2a regulated by m$^6$A methylation controls the inflammation-related negative feedback loop and cytokines in activated macrophages and alleviates heart dysfunction in the setting of sepsis.

## Results

### Spi2a is remarkably upregulated in macrophages upon lipopolysaccharide (LPS) treatment

To investigate the roles of Spi2a in macrophages, we treated macrophages with six well-characterized ligands: Pam3CSK, lipoteichoic acid (LTA), Poly(I:C), LPS, CpG, and interleukin 4 (IL-4). As shown, the expression of Spi2a in bone marrow-derived macrophages (BMDMs), peritoneal macrophages, and raw264.7 cells were largely boosted after LPS challenge compared to others (Fig. 1a and S1a, b). This finding was further confirmed in macrophages exposed to ultra-pure LPS (Fig. S1c). Moreover, LPS upregulated Spi2a expression in macrophages in a dose-dependent manner at both mRNA and protein levels (Fig. 1b, c and S1d, e), indicating Spi2a might play a critical role in activated macrophages upon LPS treatment. Spi2a transcripts were reported to be increased by NF-kappaB (NF-κB) pathway due to the κB binding site in the promotor region of Spi2a gene[40]. Next, we overexpressed IkappaB kinase β (IKKβ), a key component of IKK complex, to activate NF-κB pathway (Fig. S1f) and found the increase of Spi2a in BMDMs accordingly (Fig. 1d, e). While the addition of Bay 11-7082, a well-known NF-κB pathway inhibitor, was able to partially suppress LPS-promoted Spi2a expression in BMDMs (Fig. 1f, g), this effect on Spi2a mRNA did not change with increased dosages and showed no influences on BMDMs' viability (Fig. 1h and S1g). We then reasoned that Spi2a might be regulated in a post-transcriptional way. To this end, we performed RNA decay assays and found LPS treatment impeded Spi2a mRNA degradation in macrophages (Fig. 1i and S1h). Based on these results, we speculated that there might be post-transcriptional regulations existing on Spi2a mRNA to affect this gene's expression.

### LPS enhances METTL14 protein stability by promoting K398 acetylation of METTL14 to increase m$^6$A methylation in macrophages

Since m$^6$A methylation is the most prevalent post-transcriptional regulation in eukaryotic mRNAs[10], we next sought to detect m$^6$A methylation in LPS-treated BMDMs, peritoneal macrophages, and raw264.7 cells. As displayed, m$^6$A levels were considerably increased in macrophages upon LPS treatment (Fig. S2a). To explore the mechanism by which m$^6$A methylation is increased, we tested all of the m$^6$A-related methyltransferases and demethyltransferases. It was found that METTL14 protein expression rather than mRNA status was upregulated in macrophages after LPS challenge (Fig. S2b–d). Protein decay assays suggested that LPS facilitates METTL14 protein stability in macrophages with Cycloheximide (CHX) treatment (Fig. S2e, f). Post-translational modifications (PTMs) are reported to affect protein stability[41]. To detect the METTL14 protein's phosphorylation and acetylation modifications which are the top two most observed PTMs[41], we used a large number of proteins (40 mg) to perform co-immunoprecipitation (co-IP) against METTL14 antibody to make sure the immunoprecipitated METTL14 proteins are the same in different groups. Interestingly, METTL14 acetylation but not phosphorylation was induced by LPS in macrophages (Fig. 2a). Since lysine acetylation plays a biological role in the control of protein stability[42], we reasoned that LPS might strengthen METTL14 stability via increasing protein acetylation. Lysine acetylation is enzymatically achieved by the lysine acetylation transferases (KAT) to specifically transfer the acetyl group to lysine amino acids[25–27]. By analyzing published RNA-sequencing data (GEO: GSE153512) associated with LPS-induced BMDMs, we indicated *Kat2b* showed the most robust increase in LPS-treated BMDMs compared to PBS group among the members of Kat family (Fig. 2b). This finding was verified at both mRNA and protein levels (Fig. 2c). In addition, the recombinant KAT2B, but not KAT2A, acetylated recombinant METTL14 in an in vitro acetylation assay (Fig. 2d). L Moses

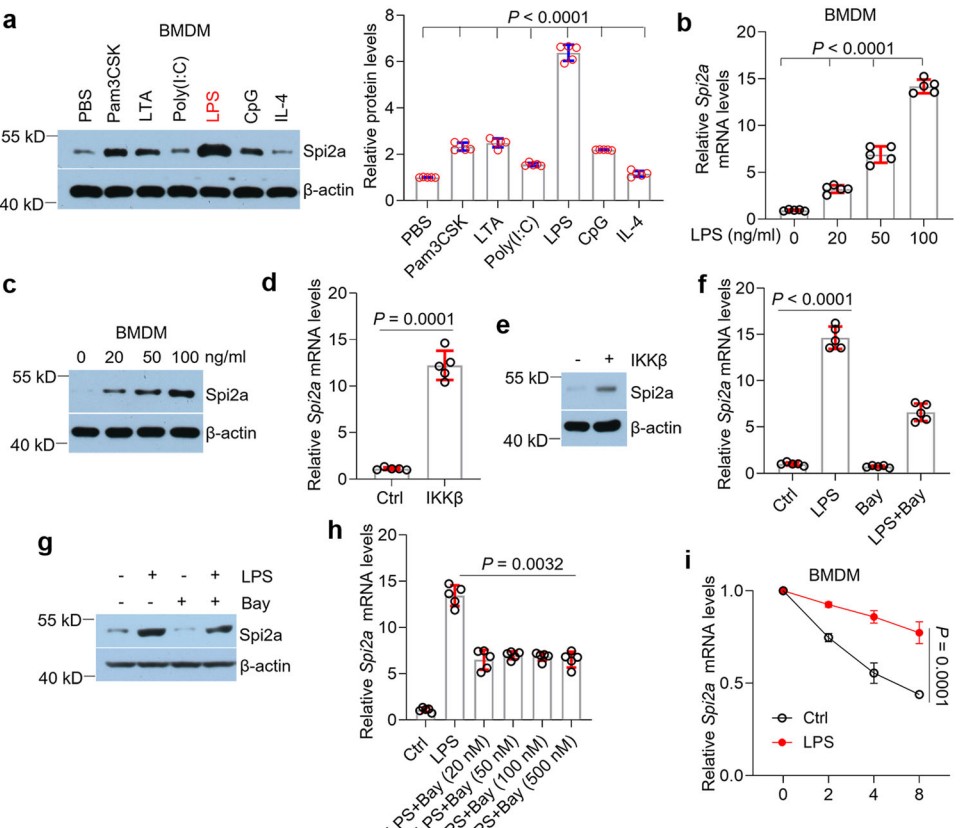

**Fig. 1 | Spi2a is upregulated in BMDMs and is increased upon LPS treatment.**
**a** Western blot (left) and quantitative analyses (right) for Spi2a protein in BMDMs with PBS, Pam3CSK, LTA, Poly(I:C), LPS, CpG, and IL-4 treatment for 8 h. **b, c** Real-time PCR (**b**) and western blot (**c**) for Spi2a in BMDMs with LPS (100 ng/ml) at different doses. **d, e** Real-time PCR (**d**) and western blot (**e**) for Spi2a in BMDMs infected with *Ikkβ*-lentivirus. **f, g** Real-time PCR (**f**) and western blot (**g**) for Spi2a in BMDMs in the presence or absence of LPS (100 ng/ml) challenge with or without 12-h Bay 11-7082 (20 nM) pre-treatment. **h** Real-time PCR for Spi2a in BMDMs in the presence of LPS (100 ng/ml) with 12-h Bay 11-7082 pre-treatment at different doses as indicated. **i** RNA decay assay of Spi2a in BMDMs with or without LPS (100 ng/ml) treatment at different time points using real-time PCR. $n = 5$ independent experiments. Cells were all treated with LPS for 8 h. Ctrl, Control; Bay, Bay 11-7082. Data are shown as mean ± SD. Unpaired two-tailed Student's *t* test (**d**), one-way (**a, b, f, h**), and two-way (**i**) two-sided ANOVA were performed for statistical analyses.

dihydrochloride, a specific KAT2B inhibitor, effectively inhibited METTL14 protein acetylation and advanced METTL14 decay in BMDMs upon LPS treatment (Fig. 2e, f). The co-IP experiments showed that KAT2B but not KAT2A (negative control) could bind METTL14 physically in BMDMs (Fig. 2g).

The removal of acetyl groups on lysine is regulated by deacetylase enzymes such as sirtuins (SIRTs) and histone deacetylases[28,29]. In macrophages, METTL14 deacetylation was confirmed to be regulated by nicotinamide (NAM) which is known as a sirtuin deacetylase inhibitor instead of the histone deacetylase inhibitor Trichostatin A (TSA) (Fig. 2h). Among the 7 members of sirtuin proteins (SIRT1-7), SIRT3, SIRT4, and SIRT5 reside in mitochondria and others play roles in nucleus and cytoplasm[29]. Thus, we purified recombinant SIRTs having activities in the nucleus and cytoplasm and recombinant METTL14 to perform in vitro deacetylation assays, which show that rSIRT1 decreased the acetylation of rMETTL14 (Fig. 2i). Forced expression of SIRT1 in BMDMs demonstrated decreased acetylation of METTL14 with either LPS or PBS treatment compared to their corresponding controls (Fig. 2j). We next generated 27 acetyl-deficient K-R mutants of mouse METTL14 according to the public protein database (https://www.ncbi.nlm.nih.gov/protein/NP_964000.2). Mutational analysis exhibited that KAT2B overexpression in BMDMs could prohibit protein decay of diverse His-tagged METTL14 mutants except K398R (Fig. S2g). Moreover, the protein stability-related acetylation site of METTL14 (K398) is conserved among different mammalian species (Fig. S2h). LPS treatment lost its effects on protein decay of acetyl-deficient K398R or acetyl-mimetic K398Q mutants (Fig. S2i). These results suggest that

LPS increases METTL14 protein levels by promoting the acetylation of K398.

## Spi2a mRNA is mediated by m6A methylation in macrophages

To explore the role of m6A methylation in Spi2a mRNA, we analyzed the published m6A-IP-sequencing data (GEO: GSE153511) using the described method[43]. Importantly, m6A methylation on Spi2a transcript was dramatically induced in M14f/f BMDMs by LPS, but this effect in M14−/− macrophages (mM14−/−) was diminished significantly (Fig. 3a). We isolated both M14f/f/ M3f/f and M14−/−/M3−/− BMDMs from mice (Fig. S3a), and then substantiated the sequencing data by m6A RIP-qPCR (Fig. 3b) as well as the m6A levels by quantification analysis, respectively (Fig. S3b). As expected, Spi2a m6A site was confirmed to interact with METTL14 and METTL3 by crosslinking-immunoprecipitation (CLIP) (Fig. 3c). Both the increase of Spi2a and enhancement of mRNA stability induced by LPS were reversed in M14−/− or M3−/− BMDMs compared to control cells (Fig. 3d and S3c). Meanwhile, CLIP assay showed decreased binding of eukaryotic translation initiation factor 3 subunit A (EIF3A) to Spi2a transcript when *Mettl14* or *Mettl3* was deleted in LPS-treated BMDMs (Fig. 3e). In newly translated protein assay, the new synthesis of L-azidohomoalanine (AHA)-labeled Spi2a was also dampened in LPS-challenged M14−/− or M3−/− BMDMs compared to the LPS control group (Fig. 3f). Together, these data claim that m6A methylation affects Spi2a mRNA stability and translation efficiency in macrophages.

METTL14 (R298P) mutant and METTL3 (D394A/W397A) mutant are reported to be defective in regulating RNA m6A modification[44].

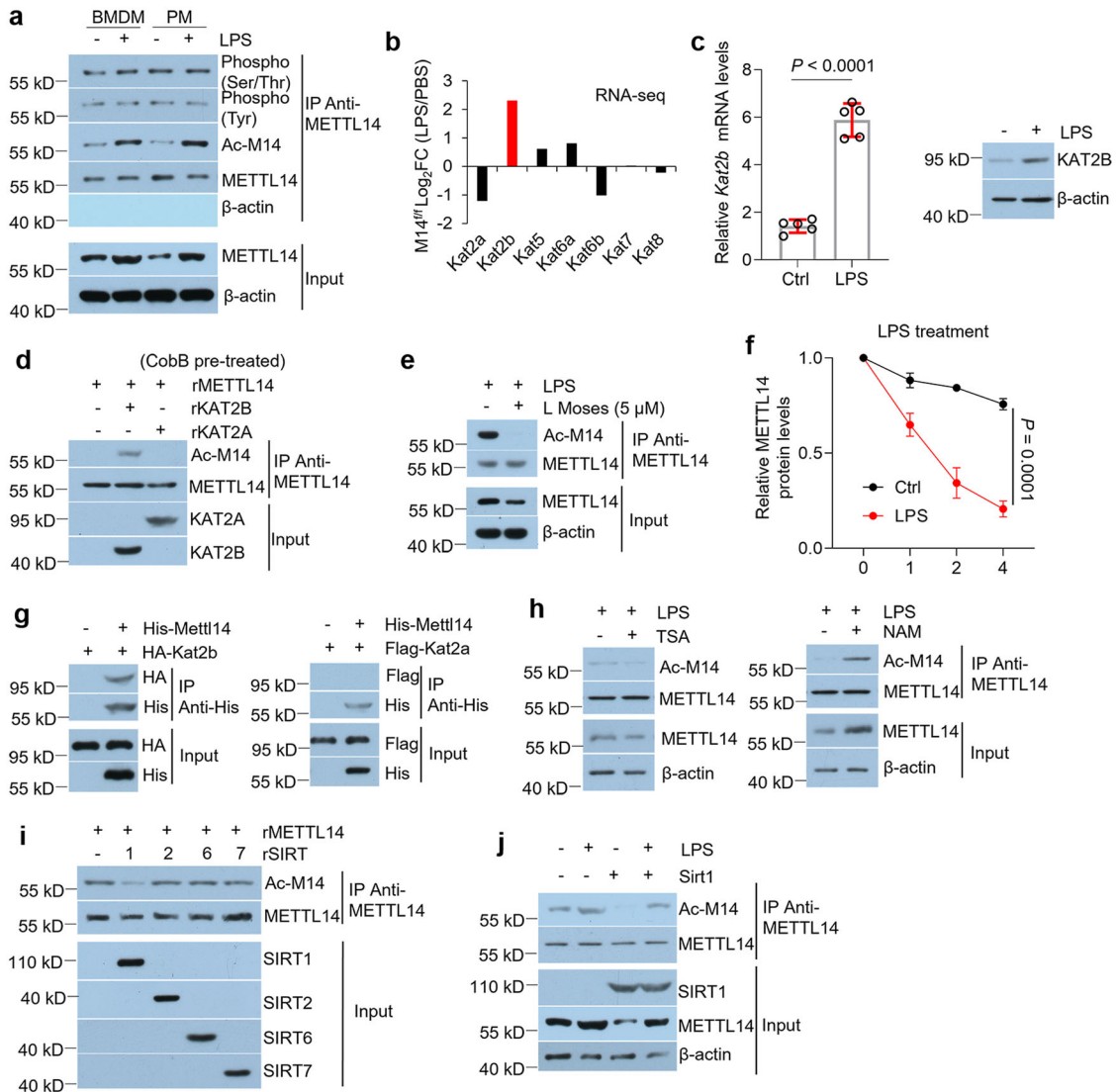

**Fig. 2 | LPS upregulates METTL14 protein levels in BMDMs. a** Western blot showing the phosphorylation and acetylation of METTL14 in BMDMs or peritoneal macrophages with or without LPS challenge. **b** RNA-sequencing data analysis of members of Kat family in M14^f/f BMDMs with or without LPS treatment. **c** Real-time PCR (left) and western blot (right) for KAT2B in BMDMs with or without LPS treatment. **d** Purified rMETTL14 were pre-treated with CobB prior to incubation with rKAT2A or rKAT2B. Immunoprecipitation (METTL14 IP) was performed and western blots were used to evaluate the acetylation of METTL14. **e** BMDMs were pre-treated with 5 μM L Moses dihydrochloride for 4 h prior to LPS treatment, co-immunoprecipitation was performed with 40 mg cell lysates, followed by western blot analysis. **f** Protein decay analyses for METTL14 in BMDMs pre-treated with 5 μM L Moses dihydrochloride for 4 h prior to LPS treatment. **g** BMDMs were infected with HA-*Kat2b*-(left), Flag-*Kat2a*-(right), or His-*Mettl14*-lentivirus as

indicated. Co-immunoprecipitation (His IP) was performed with 2 mg cell lysates, followed by western blot analyses. (**h**) BMDMs were pre-treated with or without 5 μM TSA or 10 mM NAM for 4 h, followed by LPS treatment. Co-immunoprecipitation was performed with 40 mg cell lysates, followed by a western blot to measure the acetylation of METTL14. **i** Purified rMETTL14 was incubated with rSIRT1, rSIRT2, rSIRT6 or rSIRT7. Immunoprecipitation (METTL14 IP) was performed, and a western blot was used to assess the acetylation of METTL14. **j** Ctrl- or *Sirt1*-lentivirus-infected BMDMs were treated with or without LPS. 40 mg cell lysates were subjected to co-immunoprecipitation and western blot analyses. *n* = 5 independent experiments. Ctrl, Control; Ac-M14, acetylated METTL14; r, recombinant. Data are shown as mean ± SD. Unpaired two-tailed Student's *t* test (**c**), two-tailed quasi-likelihood F test (**b**) and two-way two-sided ANOVA (**f**) were performed for statistical analyses.

Consistently, forced expression of mutants was not able to induce Spi2a in BMDMs compared to wild-type controls (Fig. S3d). To determine whether the regulation of Spi2a by METTL14 or METTL3 indeed relies on the methylation of mRNA transcript, we next generated luciferase reporter constructs with wild-type Spi2a 3' coding sequence or mutant sequence (Figure S3e). Luciferase assays suggested wild-type METTL14 or METTL3 rather than mutants promoted luciferase activity remarkably in reporters harboring WT Spi2a fragment but not mutated one (Fig. 4a), and so did LPS and KAT2B overexpression (Fig. S3f, g). To study which reader is responsible for the recognition of m6A site on Spi2a, we performed CLIP using all the readers antibodies and found that YTHDF1 interacted with m6A site on Spi2a (Fig. S3h).

We next deleted the endogenous *Ythdf1* gene in BMDMs using CRISPR/Cas 9 system (Fig. S3i). More studies on Spi2a expression, RNA decay, EIF3A-CLIP, and newly translated protein suggested that YTHDF1 regulates Spi2a expression, mRNA stability, and translation efficiency in BMDMs upon LPS challenge (Fig. 4b-d and S3j). To further confirm the interaction of YTHDF1, EIF3A, and m6A methylation site on Spi2a, we synthesized methylated single-stranded RNA (ss-m6A, with the consensus sequence GA(m6A)CC) or unmethylated RNA (ss-A) in the 3'UTR of Spi2a (Fig. 4c) to perform RNA pulldown assays in BMDMs according to previous publications[19]. As shown, YTHDF1 and EIF3A were verified to interact with the methylated RNA bait selectively with a higher affinity compared to the unmethylated control group (Fig. 4c).

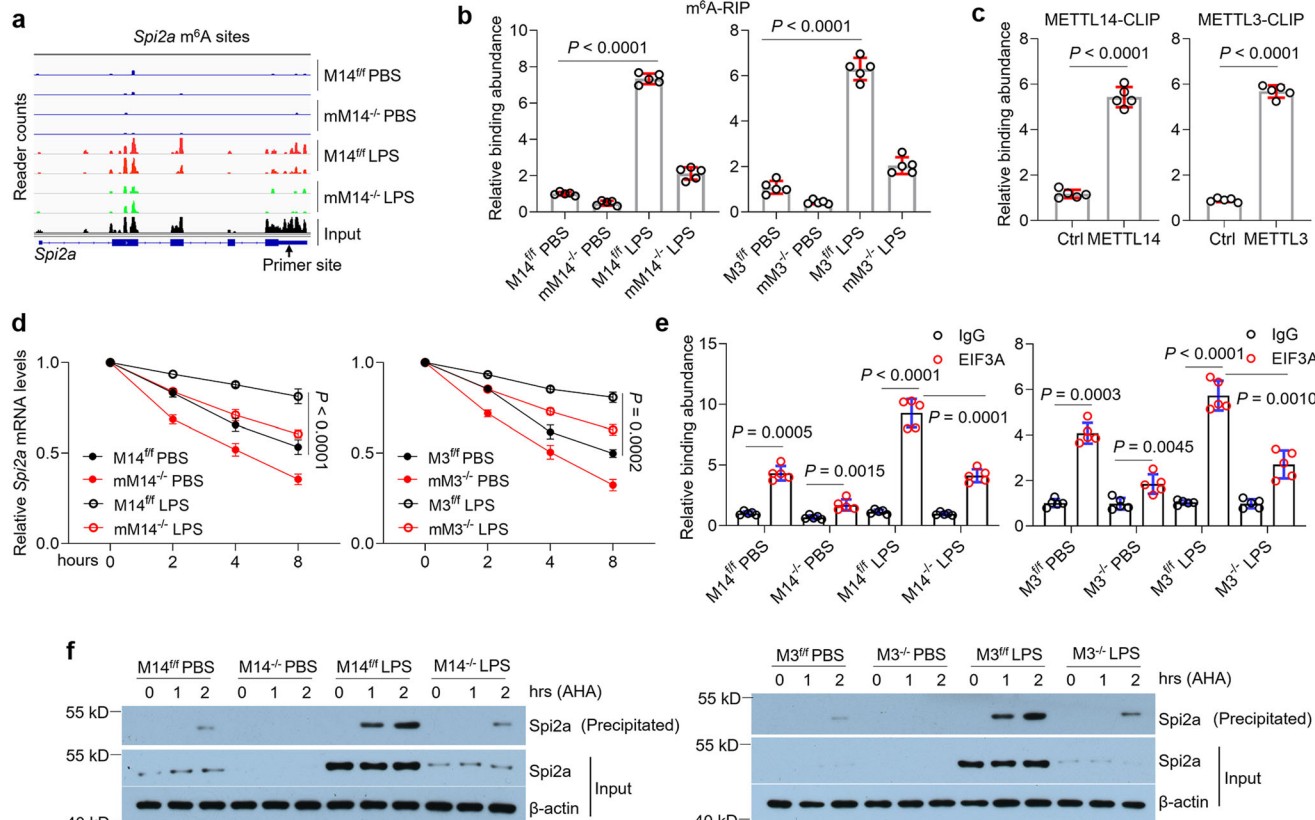

**Fig. 3 | m⁶A methylation regulates Spi2a expression. a** Read density of Spi2a transcript in M14^{f/f} and M14^{−/−} BMDMs with PBS or LPS treatment. Total RNA was isolated from BMDMs, followed by the enrichment of mRNA. mRNA was then fragmented, and 4% was used as input for internal control. The remaining 96% fragmented mRNAs mixed with anti-m⁶A antibodies were used as IP samples. Images are representative of three independent experiments. **b** RIP-qPCR assays verifying the m⁶A peaks of Spi2a transcript of m⁶A-IP sequencing data. **c** CLIP assays demonstrating the binding of METTL14 or METTL3 to the Spi2a m⁶A site in BMDMs after LPS treatment. **d** RNA decay assays for Spi2a in M14^{f/f} and M14^{−/−} BMDMs (left) or M3^{f/f} and M3^{−/−} BMDMs (right) with PBS or LPS treatment. **e** CLIP-qPCR indicating decreased association of EIF3A with Spi2a transcript in M14^{f/f} and M14^{−/−} BMDMs (left) or M3^{f/f} and M3^{−/−} BMDMs (right) with PBS or LPS treatment. **f** Western blot showing newly translated Spi2a protein labeled by L-azidohomoalanine (AHA) in M14^{f/f} and M14^{−/−} BMDMs (left) or M3^{f/f} and M3^{−/−} BMDMs (right) with PBS or LPS treatment. *n* = 5 independent experiments. Cells were all treated with 100 ng/ml LPS for 8 h. M14, Mettl14; M3, Mettl3; mM14^{−/−}, M14^{−/−} macrophages; mM3^{−/−}, M3^{−/−} macrophages. Data are shown as mean ± SD. Unpaired two-tailed Student's *t* test (**c**) and two-way two-sided ANOVA (**b**, **d**, **e**) were performed for statistical analyses.

Supported by CLIP of demethyltransferases showing FTO but not ALKBH5 bound with m⁶A site on Spi2a (Fig. 4e), we overexpressed FTO in BMDMs using lentivirus infection for further investigations (Fig. S3i). Similar to *Mettl14* deletion, reduction of m⁶A levels, a decrease of Spi2a expression, and enhancement of RNA decay were found upon LPS treatment in *Fto*-lentivirus-infected BMDMs compared to corresponding LPS control group (Fig. 4f and S3k, l). In total, these results note YTHDF1 and FTO are responsible for the m⁶A methylation of Spi2a.

**Spi2a physically interacted with IKKβ to suppress cytokines in macrophages stimulated with LPS**

To explore the roles of Spi2a in activated macrophages, we overexpressed Spi2a in macrophages (Fig. S4a), followed by LPS treatment. Interestingly, increases of cytokine mRNA transcripts triggered by LPS in macrophages were diminished by forced expression of Spi2a (Fig. 5a and S4b), so were the secretions of cytokines in the culture media (Fig. 5b and S4c). On the contrary, we deleted the Spi2a gene in macrophages using CRISPR/Cas 9 system (Fig. S4d) and found the loss of Spi2a accelerated cytokines in macrophages initiated by LPS treatment (Fig. 5c, d and S4e, f). Exposure of macrophages to LPS leads to activation of the NF-κB-dependent transcription factor, which functions to orchestrate programs of gene expression that underpin the macrophage-dependent immune reaction[45]. To this end, we tested NF-κB activity and indicated that Spi2a was able to inactivate NF-κB

pathway in macrophages exposed to LPS in the gain-of-function and loss-of-function assays (Fig. 6a and S4g). NF-κB activation is propelled by the IKK complex consisting of IKKα, β, and γ subunits which initiate IκBα degradation and trigger nuclear translocation of p65/p50 heterodimer[23,24]. Of note, Spi2a also could block the degradation of IκBα and lower IKKα/β phosphorylation induced by LPS in BMDMs (Fig. 6b). To explain the molecular basis by which Spi2a regulates NF-κB pathway, we performed co-IP using HA antibody and manifested Spi2a physically bound to IKKβ in BMDMs (Fig. 6c). Co-expression of HA-Spi2a and His-*Ikkβ* in macrophages further confirmed the interaction between Spi2a and IKKβ (Fig. 6d and S4h). The in vitro co-IP assay using recombinant proteins suggested the direct interaction between Spi2a and IKKβ (Fig. 6e). In co-IP assays against IKKγ antibody, overexpression of Spi2a could decrease the amount of precipitated phospho-IKKα/β and total IKKα/β while loss of Spi2a had an increasing effect (Fig. 6f), noting the function of Spi2a in disrupting IKK complex formation. Together, these findings demonstrate that Spi2a inhibits cytokines in LPS-activated macrophages via directly binding with IKKβ.

Supported by the analysis of RNA-sequencing data mentioned above, we noticed *Serpina3f* expression is also substantially boosted in BMDMs following LPS treatment among the 14 members of mouse Serpina3 family (Serpina3a-n) (Fig. S4i). In BMDMs exposed to LPS, we verified the upregulation of *Serpina3f* and confirmed overexpression of SERPINA3F exerted an inhibitory function in cytokines (Fig. S4i-k). However, the deletion of *Mettl14* in BMDMs did not affect *Serpina3f*

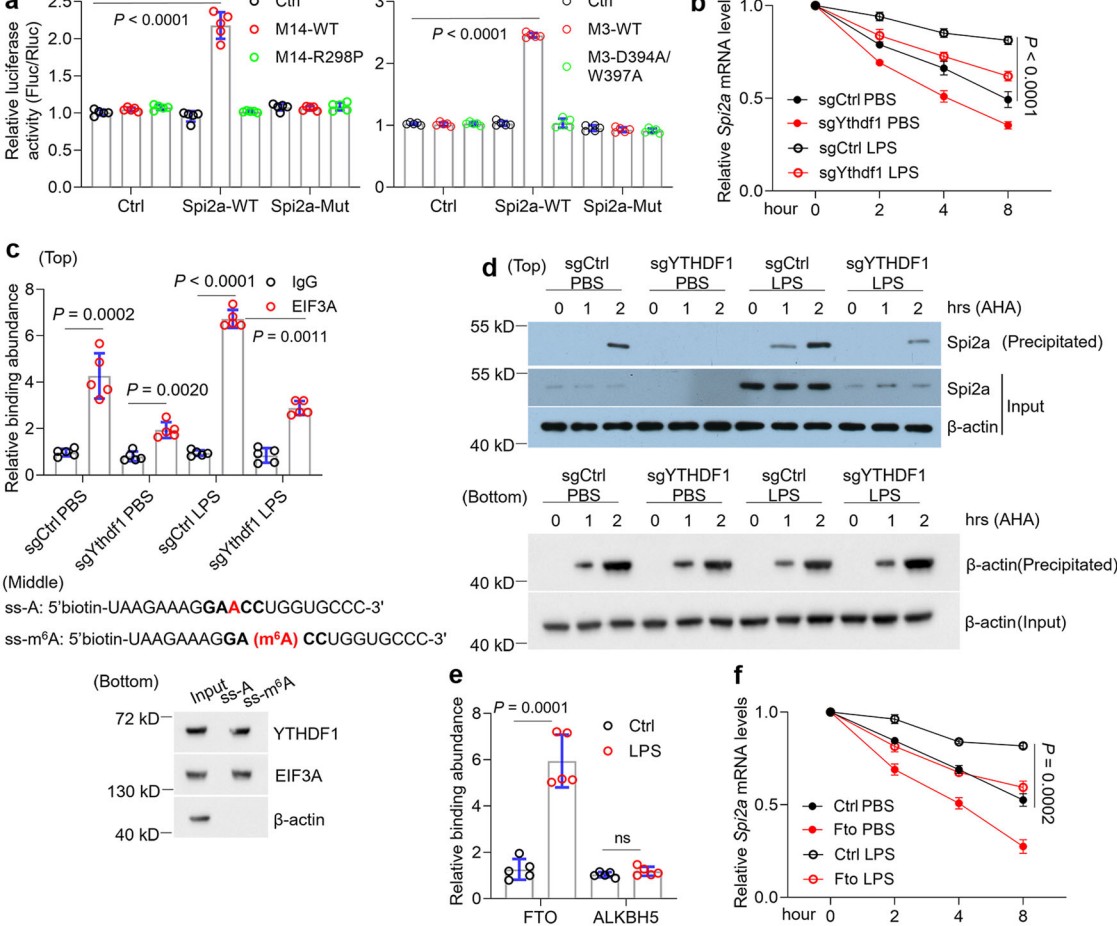

**Fig. 4 | Spi2a expression is regulated by YTHDF1 and FTO. a** Dual luciferase reporter assays exhibiting the role of METTL14 (left) or METTL3 (right) in wild-type or mutated Spi2a reporters. **b** RNA decay assays for Spi2a in control or *Ythdf1*[−/−] BMDMs with PBS or LPS treatment. **c** CLIP-qPCR indicating decreased association of EIF3A with Spi2a transcript in control or *Ythdf1*[−/−] BMDMs with PBS or LPS treatment (top); Schematic displaying ssRNA probes with unmethylated or methylated adenosine (middle); Western blotting showing the pulldown protein levels from BMDMs extract by ssRNA probes (bottom). **d** Western blot showing newly translated Spi2a and β-actin proteins labeled by L-azidohomoalanine (AHA) in control or *Ythdf1*[−/−] BMDMs with PBS or LPS treatment. Newly translated β-actin protein served as a negative control. **e** CLIP assays demonstrating the binding of FTO or ALKBH5 to the Spi2a m6A site in BMDMs with or without LPS treatment. **f** RNA decay assays for Spi2a in control or FTO-overexpressing BMDMs with PBS or LPS treatment. *n* = 5 independent experiments. Cells were all treated with 100 ng/ml LPS for 8 h. Ctrl, Control; WT, wild-type; M14, METTL14; M3, METTL3. Data are shown as mean ± SD. Unpaired two-tailed Student's *t* test (**e**) and two-way two-sided ANOVA (**a**–**c**, **f**) were performed for statistical analyses.

expression irrespective of LPS challenge (Fig. S4l), indicating *Serpina3f* mRNA is not mediated by m6A methylation.

## The loss of Spi2a in macrophages exacerbates cytokine storm and heart failure in sepsis

To study the effects of Spi2a on sepsis in animals, we depleted macrophages in wild-type mice and then reconstituted with control or Spi*2a*[−/−] BMDMs using the approach described before[43,46]. As expected, mice reconstituted with Spi*2a*[−/−] BMDMs showed 80% mortality within 96 h after LPS injection, whereas reconstitution with control BMDMs limited the mortality rate (Fig. 7a). 20 h after LPS challenge, the cytokines, and danger-associated molecular patterns (DAMPs, such as HMGB1, IL-1α and IL-33) levels in peritoneal macrophages or sera from mice reconstituted with Spi*2a*[−/−] BMDMs were higher than those in control groups (Fig. 7b, c). Myocardial dysfunction in the context of sepsis becomes the focus of intense research activities. Cardiovascular impairment in septic patients is related to a largely augmented mortality rate (70% to 90%) compared with 20% of that in sepsis without cardiovascular dysfunction[47]. To this end, we set out to examine the heart functions in this setting. As shown, hydropic change of cardiomyocytes from mice reconstituted with Spi*2a*[−/−] BMDMs was more

severe compared to that in control mice with LPS injection (Fig. 7d). Interestingly, immunohistochemistry staining indicated the appearance of F4/80-positive macrophages in the vicinity of blood vessels in heart tissues following LPS challenge but not PBS (Fig. 7e). The F4/80-positive macrophage rates in heart tissues were comparable in two kinds of reconstituted mice with LPS challenge (Fig. 7f), but more cytokines were detected in heart tissues upon Spi*2a* deletion (Fig. 7g), suggesting Spi2a plays a protective role in cytokines production rather than macrophage infiltration. Besides inflammation, TUNEL staining elucidated much more death of cardiomyocytes in mice reconstituted with Spi*2a*[−/−] BMDMs following LPS injection (Fig. S5a). Accordingly, heart function-associated factors like heart rate, left ventricular wall thickness, left ventricle ejection fraction, and left ventricular end-systolic indicated the more severe heart dysfunction of mice with Spi*2a*[−/−] BMDMs compared to that of control group with LPS (Fig. S5a). To determine whether macrophages are able to result in cardiomyocyte impairment directly, we cultured HL-1 cells supplemented with 50% (v/v) PBS or LPS-treated BMDMs' culture media. In this context, the death of HL-1 cells exposed to LPS-treated BMDMs' culture media was much more than that of control group (Fig. S5b), implying that the deleterious function of macrophage in cardiomyocytes is independent

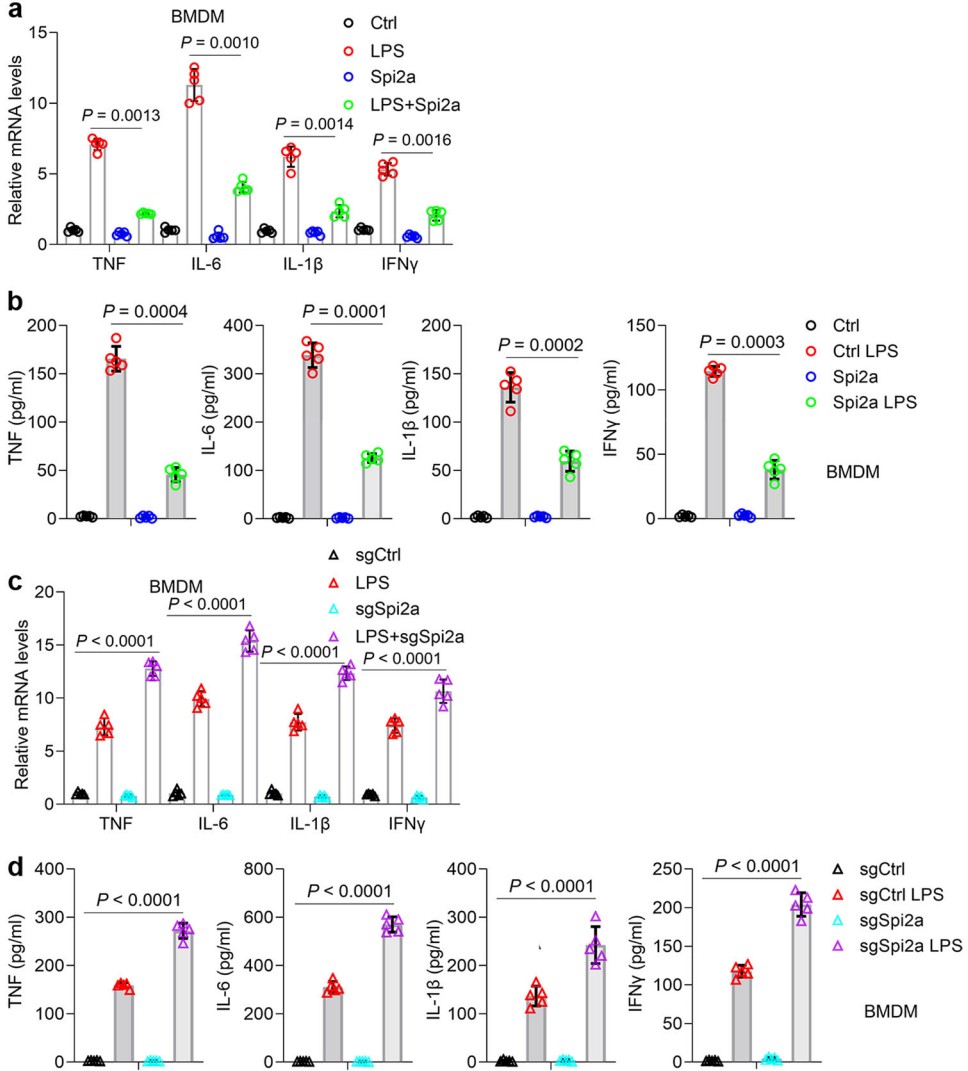

**Fig. 5 | Spi2a regulates cytokines in macrophages upon LPS treatment. a** Real-time PCR showing mRNA expression of cytokines in control- or Spi2a-lentivirus-infected BMDMs with or without LPS treatment. **b** ELISA of cytokine secretion in the culture medium of control or Spi2a-lentivirus-infected BMDMs with or without LPS treatment. **c** Real-time PCR showing mRNA expression of cytokine in sgCtrl- or sgSpi2a-lentivirus-infected BMDMs with or without LPS treatment. **d** ELISA demonstrating secretion of cytokine in the culture medium of sgCtrl- or sgSpi2a-lentivirus-infected BMDMs with or without LPS treatment. $n = 5$ independent experiments. Cells were all treated with 100 ng/ml LPS for 8 h. Ctrl, Control. Data are shown as mean ± SD. Two-way two-sided ANOVA were performed for statistical analyses.

of other immune cells. Cecum ligation and puncture (CLP) model resembling many aspects of human sepsis is another widely utilized animal model of sepsis[48]. Similar to LPS model, the loss of Spi2a showed a negative effect on cytokine storm, DAMPs, and heart failure in CLP animal model (Fig. S5c–j).

## The loss of m⁶A methylation fosters cytokine storm and heart dysfunction in sepsis

We have confirmed Spi2a is tuned by m⁶A methylation in macrophage above. To elucidate whether m⁶A methylation of macrophages is involved in sepsis development, we treated M14^f/f and mM14^−/− mice with or without LPS. As shown, mortality rate, inflammatory response, and heart dysfunction were all aggravated in mM14^−/− mice compared to M14^f/f mice following LPS injection, so were them in M3^f/f and mM3^−/− mice (Fig. S6a–n). These findings were also confirmed in CLP animal models (Fig. S6o–z). Since Lyz2^Cre is also expressed in neutrophils[49], we depleted macrophages in M14^f/f and mM14^−/− mice and then reconstituted them with fully differentiated M14^f/f and M14^−/− BMDMs crossly (M14^f/f BMDM > mM14^−/− mice; M14^−/− BMDM > M14^f/f mice) to

demonstrate that macrophages are the cause of septic response. Similarly, in M14^f/f recipients transferred with M14^−/− BMDMs, the severity of mortality rate, inflammatory response and heart dysfunction were higher than those of control group under septic conditions (Fig. S7). To confirm the roles of YTHDF1 and FTO in vivo, we depleted macrophages in wild-type mice and then reconstituted with or without Ythdf1^−/−/FTO-overexpressed BMDMs. As shown, either deletion of Ythdf1 or forced expression of FTO in macrophages caused more mortality, boosted inflammatory response, and worse heart dysfunction in animal sepsis models (Fig. S8).

## Overexpression of Spi2a in macrophages ameliorates inflammatory response and myocardial dysfunction in sepsis

To investigate whether forced expression of Spi2a could counteract septic response, we depleted macrophages in wild-type mice and then reconstituted with or without BMDMs overexpressing Spi2a. As anticipated, mortality was decreased to 10% in the setting of Spi2a overexpression in macrophages compared to 30% of that in control mice in response to LPS (Fig. 8a). Meanwhile, cytokines or DAMPs

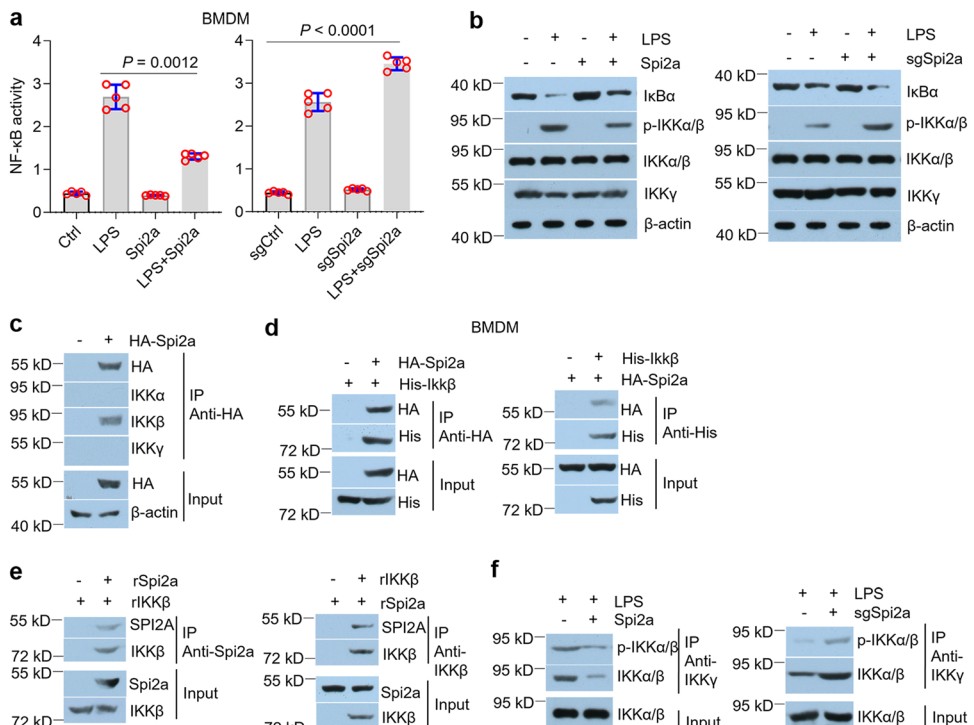

**Fig. 6 | Spi2a binds IKKβ in BMDMs to suppress cytokines. a** NF-κB activity assays of control- or Spi2a-lentivirus-infected BMDMs (left) or sgCtrl- or sgSpi2a-lentivirus-infected BMDMs (right) with PBS or LPS challenge. **b** Western blot analyses of control- or Spi2a-lentivirus-infected BMDMs (left) or sgCtrl- or sgSpi2a-lentivirus-infected BMDMs (right) with PBS or LPS challenge. **c** BMDMs were transduced with control or HA-Spi2a-lentivirus and immunoprecipitated with HA antibody, followed by western blot analyses. **d** BMDMs were infected by HA-Spi2a- or His-Ikkβ-lentivirus as indicated and immunoprecipitated by HA (left) or His (right) antibody,

followed by western blot analyses. **e** Recombinant Spi2a and IKKβ were mixed and immunoprecipitated by Spi2a (left) or IKKβ (right) antibody as indicated, followed by western blot analyses. **f** BMDMs were infected with Spi2a- (left) or sgSpi2a-lentivirus (right) as indicated prior to LPS treatment. Co-IP assays were performed using IKKγ antibody, followed by western blot analyses. $n = 5$ independent experiments. Cells were all treated with 100 ng/ml LPS for 8 h. Ctrl, Control. Data are shown as mean ± SD. Two-way two-sided ANOVA were performed for statistical analyses.

levels in peritoneal macrophages, sera and heart tissues from mice with BMDMs overexpressing Spi2a were largely relieved in contrast to control mice with LPS (Fig. 8b–d), and so were hydropic degeneration of cardiomyocytes, cell death of heart tissues and heart dysfunction (Fig. 8e and S9a, b). We further questioned whether forced expression of Spi2a could offset septic response in mM14[-/-] mice upon LPS challenge. To accomplish this, we depleted macrophages in wild-type mice, and then reconstituted them with M14[f/f] or M14[-/-] BMDMs infected with or without Spi2a-lentivirus. Consistently, compared to corresponding controls, forced expression of Spi2a markedly improved the survival (Fig. 8f), restricted pro-inflammatory cytokines and DAMPs, and corrected myocardial dysfunction in mice reconstituted with M14[-/-] BMDMs under LPS challenge condition (Fig. S9c–e). These results were further confirmed in the CLP animal model (S9f–q).

### Human orthologue SERPINA3 suppresses LPS-induced cytokines in human macrophages

To decipher whether Spi2a could compromise cytokines in human macrophages, we overexpressed Spi2a in both monocyte-derived macrophages and THP-1 cells. As shown, forced expression of Spi2a restrained cytokine generation in human macrophages in response to LPS (Fig. S10a, b). Moreover, SERPINA3, the human orthologue of Spi2a, was also upregulated in the presence of LPS (Fig. S10c). Accompanied by the changes of m6A levels, SERPINA3 expression was regulated by METTL14 in human macrophages (Fig. S10d, e). We next analyzed the amino acid sequences between Spi2a and SERPINA3 using the online software Uniprot (https://www.uniprot.org/blast/). The percent identity of them is 58.5% ($P = 6.9 \times 10^{-154}$), suggesting they might exert similar biological functions. Indeed, overexpression of

SERPINA3 limited LPS-induced cytokine production and NF-κB activation in human macrophages (Fig. 9a, b, S10f). KAT2B, which was fostered by LPS (Fig. S10g), could enhance SERPINA3 expression at both mRNA and protein levels (Fig. S10h, i). Like the data in BMDMs, SERPINA3 interacted with IKK β physically in human macrophages (Fig. S10j), indicating both Spi2a and SERPINA3 share the same functional amino acids for IKK complex disruption. While SIRT or FTO inhibitor treatment, which contributes to elevation of m6A methylation, relieved cytokine production in response to LPS, the combination of them was more efficient (Fig. S10k, l). Suppressor of cytokine signaling 1 (SOCS1) is reported to be essential for maintaining the negative feedback control of activated macrophages[43]. Of note, cytokine levels in human macrophages with LPS challenge were dropped near to the baseline with SERPINA3- and SOCS1-lentivirus co-infection compared to SOCS1 alone group (Fig. 9a, b, S10m), which highlights that combination drug therapy on multiple targets might be a better approach for sepsis management in clinic.

### Upregulation of SERPINA3 in monocytes is important for cytokines inhibition in sepsis patients

To interrogate the roles of SERPINA3 in clinic, we collected peripheral blood samples and enriched monocytes from 15 sepsis patients and 15 healthy volunteers. As shown, LPS concentrations in serums derived from patients were much higher than those in healthy controls (Fig. 10a). Accordingly, m6A levels, SERPINA3, and cytokines expression were all boosted in diseased specimens (Fig. 10b, c). In monocytes of sepsis, the status of SERPINA3 being positively correlated with m6A levels showed a negative correlation with cytokines expression (Fig. 10d–h). Moreover, SERPINA3 and m6A levels in monocytes were negatively correlated with the clinical parameters of inflammation and

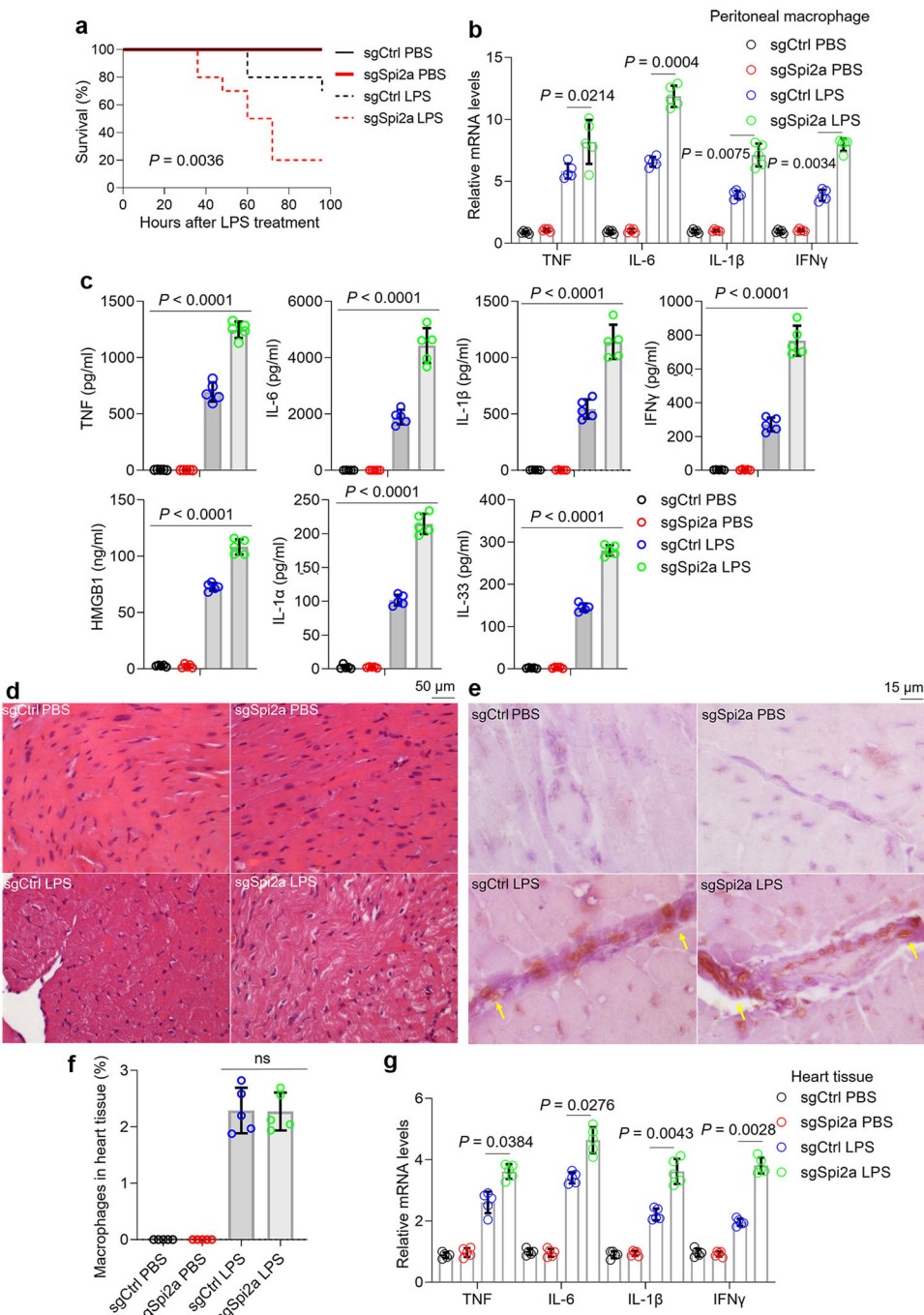

**Fig. 7 | Effect of Spi2a depletion in cytokine storm and heart dysfunction in animal sepsis models.** Macrophage-depleted wild-type mice were reconstituted with wild-type BMDMs transduced with sgCtrl or sg*Spi2a* lentivirus prior to LPS challenge. **a** Kaplan–Meier survival curves. *n* = 10 in each group. **b** Real-time PCR for cytokine expression in peritoneal macrophages. *n* = 5 in each group. **c** ELISA for TNF, IL-6, IL-1β, IFNγ, HMGB1, IL-1α and IL-33 in serum. *n* = 5 in each group. **d** HE examination of heart tissues. Imageaware representative of 5 independent experiments. **e** IHC staining for F4/80+ macrophages (indicated by yellow arrows) in heart tissue. Images are representative of 5 independent experiments. **f** Magnetic-activated cell sorting for the percentage of F4/80+ macrophages in heart tissue. *n* = 5 in each group. **g** Real-time PCR for cytokine expression in heart tissue. n = 5 in each group. Ctrl, Control; mM14−/−, M14−/− macrophages. Data are shown as mean ± SD. Log-rank test (**a**) and two-way two-sided ANOVA (**b**, **c**, **f**, **g**) were used for statistical analyses.

myocardial dysfunction such as C-Reactive Protein (CRP) and Troponin T, High Sensitivity (hs-TnT) (Fig. 10i–l), indicating the protective functions of SERPINA3 and RNA methylation of blood monocytes in sepsis.

## Discussion

Sepsis is considered to be a life-threatening disease characterized by overwhelming systemic inflammation as well as organ dysfunction owing to the dysregulated host response upon infection[7].

Macrophages, the key effector for innate immunity, take an essential effect on host defense against microorganism infection[9]. Upon infection, bacteria trigger inflammatory response via the Toll-like receptor 4 (TLR4)/NF-κB signaling pathway in macrophages[45,50]. In addition, there are several negative intracellular regulators such as SOCS1, suppression of tumorigenicity 2 protein (ST2), and A20 induced by LPS through a negative feedback mechanism, all of which play critical roles in controlling cytokines status in activated macrophages[51–53]. In this

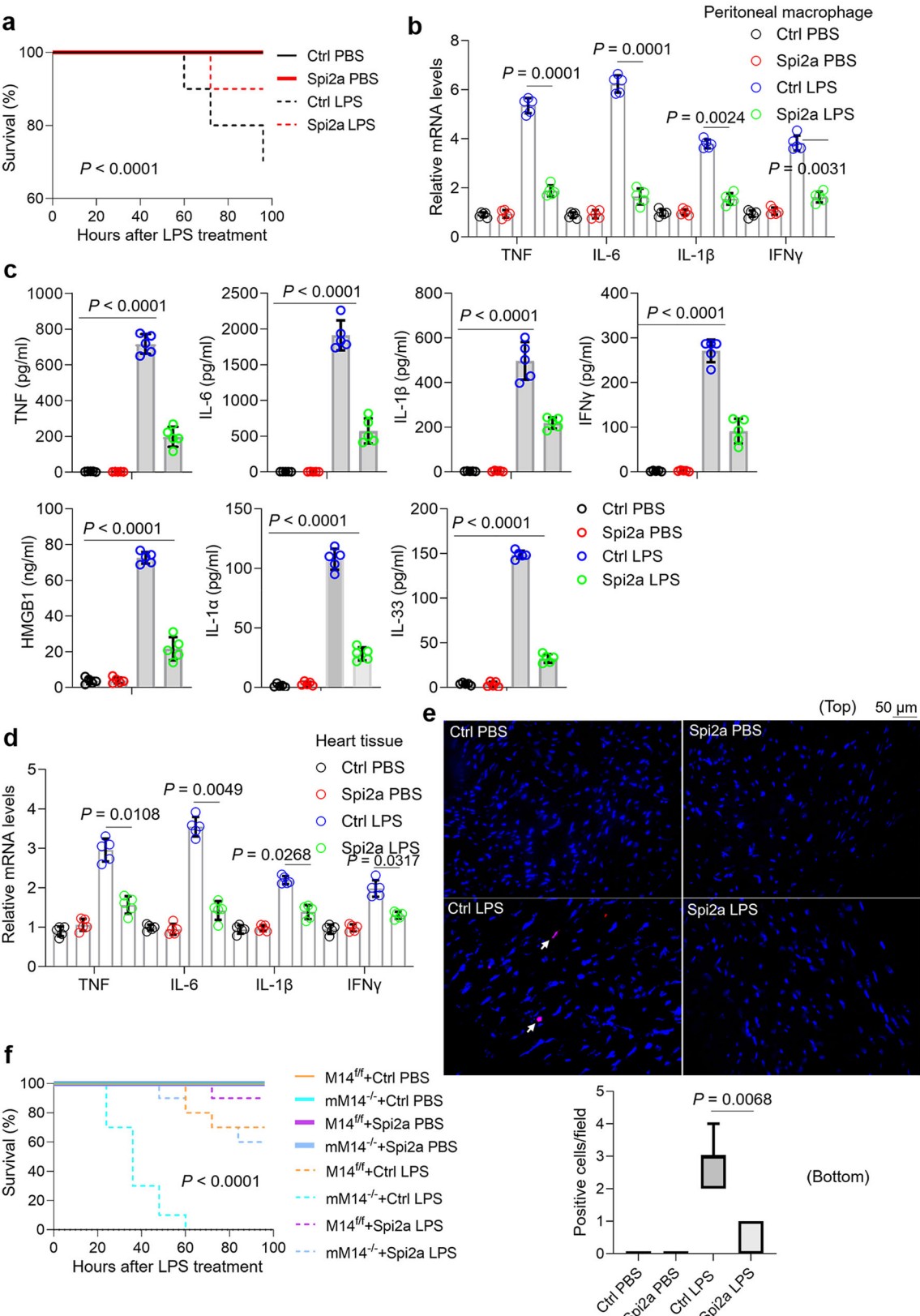

study, we identified a novel negative feedback regulator Spi2a in mice (SERPINA3 in humans), which acted to suppress cytokines in macrophages following LPS challenge. This is accomplished through curbing IKK complex formation and inhibiting NF-κB activation, which is consistent with other studies demonstrating Spi2a or SERPINA3 is closely associated with immune reactions[36,54,55].

Here we found that LPS is able to increase METTL14 levels. In line with recent investigations noting that LPS upregulates RNA methylation through degrading *Fto* mRNA[43], we also found FTO protein levels are clearly reduced in macrophages upon LPS treatment. Therefore, both METTL14 activation and FTO reduction work together to increase m6A levels upon bacterial infection or LPS treatment.

**Fig. 8 | Effect of Spi2a overexpression in cytokine storm and heart dysfunction in animal sepsis models.** Macrophage-depleted wild-type mice were reconstituted with wild-type BMDMs transduced with control- or Spi2a-lentivirus prior to LPS challenge. **a** Kaplan–Meier survival curves. *n* = 10 in each group. **b** Real-time PCR for cytokine expression in peritoneal macrophages. *n* = 5 in each group. **c** ELISA for TNF, IL-6, IL-1β, IFNγ, HMGB1, IL-1α and IL-33 in serum. *n* = 5 in each group. **d** Real-time PCR for cytokine expression in heart tissue samples. *n* = 5 in each group. **e** TUNEL staining for dead cells (indicated by white arrows) detection in heart tissues (top) and quantitative analysis of TUNEL staining-positive cells in heart

tissues displayed as box-and-whiskers plots (bottom). *n* = 20 fields of view in each group. **f** Macrophage-depleted wild-type mice were reconstituted with M14^f/f or M14^−/− BMDMs transduced with control- or Spi2a-lentivirus prior to LPS challenge, then survival experiments were performed, *n* = 10 in each group. Ctrl, Control; mM14^−/−, M14^−/− macrophages. Data are shown as mean ± SD. For box-and-whisker plots, the bottom line shows the lower quartile, top line the upper quartile, and whiskers the maxima/minima. Log-rank tests (**a**, **f**) and two-way two-sided ANOVA (**b**–**e**) were used for statistical analyses.

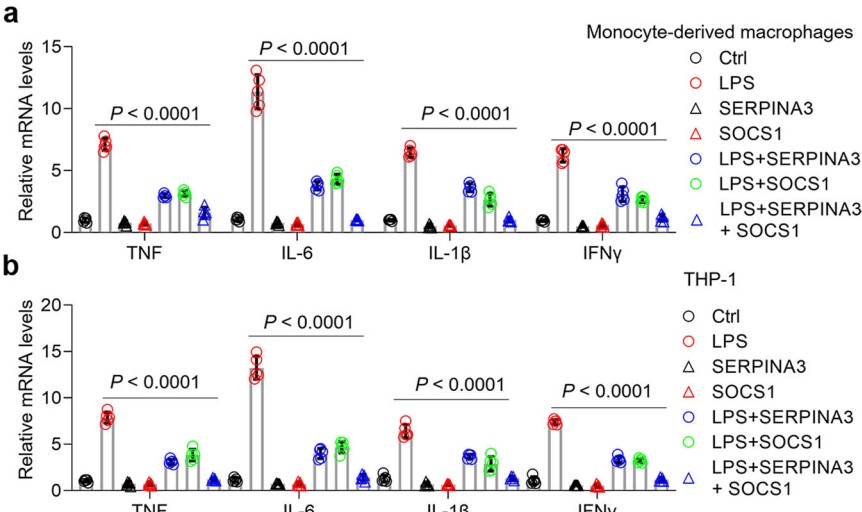

**Fig. 9 | Overexpression of SERPINA3 in human macrophages rescues overwhelming cytokines in response to LPS. a, b** Real-time PCR analyses demonstrating cytokine transcripts in PBS- or LPS-treated monocyte-derived

macrophages (**a**) or THP-1 cells (**b**) with or without *SERPINA3*- or *SOCS1*-lentivirus infection as indicated. Ctrl, Control. Data are shown as mean ± SD. Two-way two-sided ANOVA was performed for statistical analyses.

Lysine acetylation, an important post-translational modification, influences proteins' stability, localization, and function. Despite originally thought to occur on histones only, numerous nonhistone proteins are reported to be acetylated now. In this acetylated event, KATs are involved in protein complexes that carry out modifications[28]. Our data exhibited that LPS regulates METTL14 protein, but not *Mettl14* mRNA, stability. Moreover, we confirmed LPS enhanced METTL14 protein stability via inducing lysine acetylation transferase KAT2B to elevate m6A methylation in macrophages, suggesting the multifaceted functions of LPS in altering RNA methylation. Sirtuins can deacetylate histones and nonhistone targets in mice and humans[28]. Here, we reported that SIRT1 is responsible for the deacetylation of the METTL14 protein. Therefore, KAT2B and SIRT1 account for the dynamic regulation of METTL14 acetylation. Due to the lack of a specific anti-acetylated lysine antibody against METTL14, we immunoprecipitated METTL14 protein and then detected the acetylation levels of this protein using a commercial anti-acetylated lysine antibody.

We provided compelling evidence that m6A methylation could regulate Spi2a mRNA decay and translation efficiency. The loss of RNA methylation in macrophages aggravated cytokine storm and myocardial dysfunction, in line with the study indicating that m6A methylation is required to maintain the negative feedback control of activated macrophage[43]. Macrophage depletion and reconstitution system is a powerful tool for macrophage-related research[46]. In this system, mice were injected with clodronate to eliminate the endogenous macrophages and reconstituted with Spi2a-depleted macrophages. Upon Spi2a depletion, enhanced cytokine storm and myocardial damage were shown in mice upon LPS challenge. Moreover, using this tool, we overexpressed Spi2a in macrophages and found forced expression of Spi2a rescues the hyper-responsive phenotype in mice.

In addition to its role in macrophages as we explored here, SERPINA3 is evidently increased in the SARS-CoV-2-infected respiratory cells of patients[56], indicating its potential effects on inhibiting viral replication of SARS-CoV-2. Therefore, SERPINA3 is a candidate target for the control of both sepsis and COVID-19. Based on our data, the combination of SIRT1 and FTO inhibitors which contribute to m6A methylation increases might be an effective therapeutic approach for the prevention or management of sepsis and COVID-19 in clinic.

## Methods
### Animals
Wild-type C57BL/6 (Cat#: SM-001), *Mettl14*^flox/flox (Cat#: NM-CKO-190007), and *Mettl3*^flox/flox (Cat#: NM-CKO-190006) mice bearing LoxP sites were from Shanghai Model Organisms Center. LysM-Cre transgenic mice (Cat#: 004781) were from Jackson Laboratory. *Mettl14*^flox/flox/*Mettl3*^flox/flox mice and LysM-Cre transgenic mice were crossed to generate the conditional gene knockout animals. All mice were of the C57BL/6 background, aged 6–8 weeks, and were used in this study irrespective of sex. The mice were housed at 25 °C with a 12 h/12 h light/dark cycle and ~50% humidity. The experimental and control mice were bred separately under specific pathogen-free conditions. For experiments, the mice were euthanized in a CO2 chamber. The protocols related to animals were approved by the Institutional Animal Care and Use Committee (IACUC) of Shanxi Medical University. Both male and female mice were used as our preliminary data did not show sex as a confounding factor.

### Cell lines
THP-1 (TIB-202), RAW 264.7 (TIB-71), L929 cell lines (CCL-1), and HEK293T cells (CRL-3216) were purchased from ATCC. Human THP-1 cells were plated in RPMI 1640 medium supplemented with

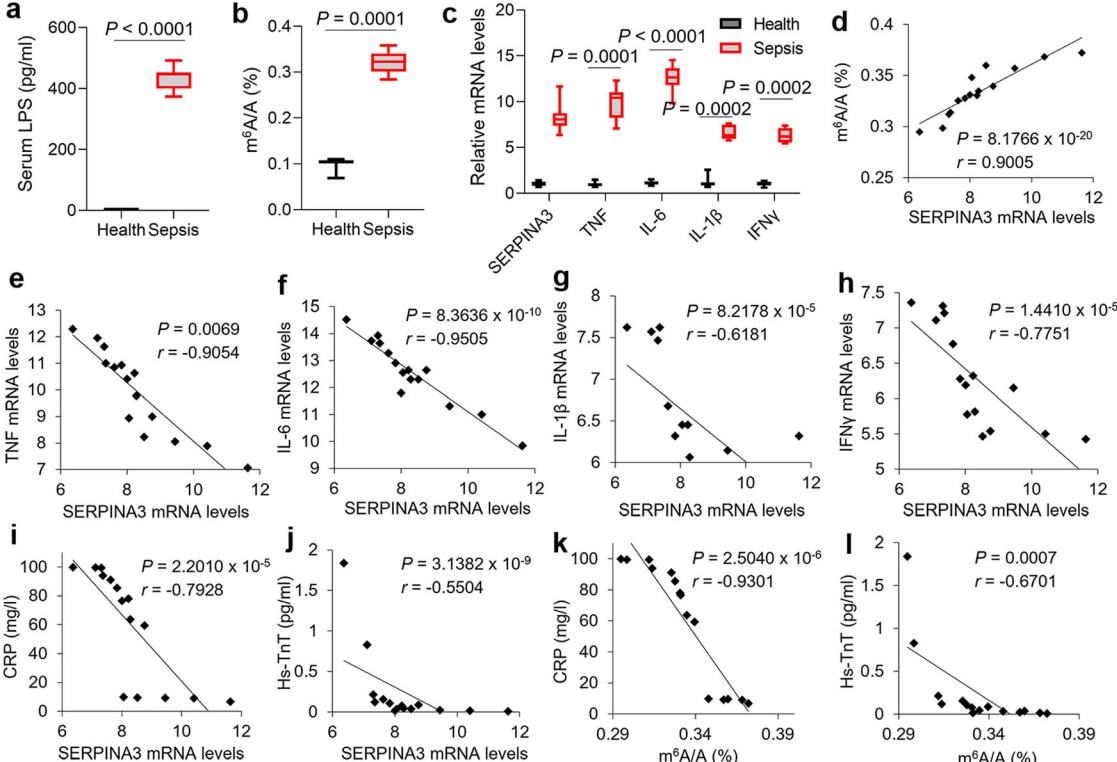

**Fig. 10 | The roles of SERPINA3 in patients. a** Box-and-whisker plot showing the LPS concentrations in the serums from healthy individuals and sepsis patients, $n = 15$ independent samples for each group. **b** Quantification analysis of m⁶A levels in the blood monocytes of healthy individuals and sepsis patients displayed by box-and-whisker plot, $n = 15$ independent samples for each group. **c** Box-and-whisker plot exhibiting SERPINA3 and cytokine mRNAs expression in the blood monocytes of healthy individuals and sepsis patients, $n = 15$ independent samples for each group. **d**–**h** Pearson's correlation analysis among m⁶A levels, SERPINA3, TNF, IL-6, IL-1β and IFNγ in the peripheral blood monocytes from sepsis patients. **i, j** Pearson's correlation analysis between SERPINA3 expression in the peripheral blood monocytes and clinical parameters of sepsis patients. **k, l** Pearson's correlation analysis between m⁶A levels in the peripheral blood monocytes and clinical parameters of sepsis patients. Data are shown as mean ± SD. For box-and-whisker plots, the bottom line shows the lower quartile, top line the upper quartile, and whiskers the maxima/minima. Unpaired two-tailed Student's *t* tests (**a**–**c**) and Pearson's correlation (**d**–**l**) were used for statistical analysis.

L-glutamine (2 mM), 10% FBS, HEPES (10 mM, pH 7.2), β-mercaptoethanol (0.05 mM), and 1% penicillin–streptomycin. THP-1 cells were treated with phorbol 12-myristate 13-acetate (PMA, 100 ng/ml, Sigma-Aldrich, Cat#: P8139) for 48 h to differentiate into macrophages. RAW 264.7 mouse macrophages and HEK293T cells were cultured in DMEM containing 10% FBS and 1% penicillin–streptomycin at 37 °C and 5% $CO_2$. L929 mouse fibroblast line was grown in DMEM containing 10% FBS and 1% P/S at 37 °C and 5% $CO_2$. Conditioned media were collected at day 3 after confluency, filtered, and stored in a −20 °C freezer. BMDMs, peritoneal macrophages, and raw264.7 cells were treated with Pam3CSK4 (100 ng/ml, Invitrogen, Cat#: tlrl-pms), LTA (100 ng/ml, Invitrogen, Cat#: vtlrl-lta), Poly(I:C) (50 µg/ml, Sigma-Aldrich, Cat#: P1530), LPS (100 ng/ml, Sigma-Aldrich, Cat#: L2630), CpG (1 µM, Invitrogen, Cat#: tlrl-1668) or IL-4 (100 ng/ml, Sigma-Aldrich, Cat#: SRP3211) for 8 h.

### Human blood monocytes isolation and differentiation
Fresh human blood was harvested from healthy donors according to the procedure approved by the Ethical Committee on Human Research of Shanxi Medical University. Informed consent was received from all individuals. Blood mononuclear cells were enriched using Ficoll-Paque density gradient cell separation (GE Healthcare) based on the manufacturer's protocol. The layer with mononuclear cells was collected, re-dissolved in 25 ml cold PBS, and then centrifuged for 10 min at 350 × *g*. Pan Monocyte Isolation Kit (Miltenyi Biotech) was applied to further purify monocytes from blood mononuclear cells. In some experiments, purified monocytes were plated in RPMI 1640 containing 1% P/S, 10% FBS, glutamine (2 mM), 1% non-essential amino acids, 1%

sodium pyruvate and human recombinant M-CSF (20 ng/ml, Pepro-Tech, Cat#: 300-25) for 7 days to differentiate monocytes toward macrophages.

### Human sepsis samples
Human peripheral blood samples from 15 sepsis patients and the corresponding 15 healthy individuals were from Hainan General Hospital. The consensus conference definition[4] was applied for the inclusion criteria of sepsis which was further verified with blood culture for *Escherichia coli*. The exclusion criteria for sepsis were: the presence of chronic inflammatory diseases or malignancy, treatment with immunosuppressive drugs or steroids in one month, HIV/AIDS, hepatic failure, hepatitis B or C, and pregnancy. This procedure was approved by the Ethical Committee on Human Research of Hainan General Hospital (#Med-Eth-Re[2022]515). Informed consent was received from all individuals. Informed consent to publish information about individuals were received. Sex was considered in the study design and sex of participants was determined based on self-report. The ratio of male and female for each group is around 1:1. The sex and ages between healthy individuals and patients were matched. Blood mononuclear cells were enriched as described above. More details on patients were listed in Supplementary Table 2. Analyses of clinical parameters in Supplementary Table 3 suggested sex was not a confounding factor.

### Macrophage isolation and culture
Macrophages in the peritoneal cavity were normally isolated by injecting 5 ml cold PBS (pH 7.0) intraperitoneally and collecting the

PBS fluids. For the primary peritoneal macrophages culture, thioglycollate was used to enrich macrophages in the peritoneal cavity as described previously[57] and peritoneal macrophages were cultured in RPMI 1640 media with heat-inactivated FBS (10%), penicillin (100 U/ml) and streptomycin (100 mg/ml). BMDMs were isolated according to published studies with minor modifications[43]. In brief, bone marrow cells were flushed out of the legs of mouse using cold PBS containing 1% FBS, followed by the elimination of red blood cells with NH$_4$Cl (10 mM, pH8.0). The isolated bone marrow cells were cultured in RPMI 1640 medium for 5 h and the suspended cells were replated in a new petri dish with differentiation medium (70% RPMI 1640 culture medium + 30% L929 conditioned media). The replated cells would differentiate into BMDMs in 7 days. In some experiments, the suspended cells were treated with a lentivirus and polybrene (6 mg/ml) for 48 h prior to differentiation medium treatment.

### Co-immunoprecipitation (Co-IP)

Co-IP assays were performed based on published investigations[58]. In brief, infected- or treated-macrophages were lysed in IP lysis buffer (150 mM NaCl, 25 mM Tris-HCl (pH 7.4), protease inhibitor, and 1% NP-40) and supernatants were harvested by centrifugation. 10% of cell lysates (2-40 mg) was saved as input and the other 90% was incubated with protein A/G beads (ThermoFisher Scientific, Cat#: 88802) and antibodies. After 4-h rotation, protein-antibody-bead mixtures were washed with NETN buffer (100 mM NaCl, 20 mM Tris-Cl (pH 8.0), 0.5 mM EDTA and 0.5% NP-40) and subjected to western blot analysis. For recombinant protein interaction, total proteins were incubated with antibodies for 4 h and then with protein A/G beads for another 2 h at 4 °C prior to western blot analysis.

### Prokaryotic recombinant proteins purification

High-expressing BL21(DE3) pLysS cells were grown in a 250 mL culture and subjected to IPTG-induction at 30 °C for 16 h. His-tagged proteins were enriched through sonication of bacteria cells. Cell lysates were collected and subjected to a Ni-NTA column within imidazole (20 mM). The proteins bound to the column were eluted with imidazole (250 mM), and then desalted by a PD-10 column. The purified proteins were verified by western blot analysis and Coomassie Brilliant Blue staining.

### In vitro acetylation and deacetylation assays

Purified recombinant METTL14 protein was incubated with recombinant KAT2A/KAT2B acetyltransferase (100 ng) in 30 mL reaction buffer (75 mM potassium chloride (KCl), 40 mM Tris-HCL (pH8.0) and 10 mM acetyl CoA) at 30 °C for 45 min. For deacetylation experiment, 2 mg of purified recombinant METTL14 was incubated with 300 ng SIRT1, SIRT2, SIRT6, or SIRT7 in the SDAC buffer (0.5 mM DTT, 4 mM MgCl$_2$, 50 mM NaCl, 50 mM Tris-HCL, 1 mM NAD$^+$, 0.5 μM TSA) for 3 h at 30 °C with rotation. SDS–PAGE sample buffer was added to terminate acetylation reaction and protein samples were subjected to western blot analysis.

### CobB treatment

In the deacetylation assay, CobB (5 mg) was mixed with 10 mg of rMETTL14, rKAT2A, or rKAT2B in deacetylation reaction solution (Tris-HCL (50 mM, pH 8.0), NaCl (135 mM), KCl (2.5 mM) and MgCl2 (1 mM) to delete intrinsic lysine acetylation derived from bacteria.

### RT-PCR

Total RNAs in fresh tissues or cells were lysed and extracted by Trizol Reagent (Invitrogen, Cat#: 15596026). The reverse transcription was performed using a qPCR RT kit (TOYOBO, Cat#: FSQ-101). Real-time PCR was performed by an SYBR Green Real-time PCR Master Mix kit (TOYOBO, Cat#: QPK-201) in a real-time PCR system (Roche). The 2$^{-\Delta\Delta Ct}$ formula was used to calculate the relative amounts of transcripts.

GAPDH served as an internal control. Primers were listed in Supplementary Table 1. Real-time PCR data were collected by the Roche 480 II system and analyzed by Microsoft Excel 2013.

### Western blot

Cells or tissues were lysed in RIPA buffer containing 5% 2-mercaptoethanol and Protease Inhibitor Cocktail on ice for 5 min, and denatured at 95 °C for 10 min. Protein concentration of each sample was measured by the BCA Protein Assay Kit. Proteins in lysates were separated by SDS–PAGE and then transferred onto 0.45 μm polyvinylidene fluoride membranes. The membranes were blocked in 2% BSA for 1 h at room temperature, followed by overnight incubation with primary antibodies (1:1000 dilution) in cold room. On day 2, membranes were washed by TBST buffer and incubated with anti-mouse or anti-rabbit IgG-horseradish peroxidase(HRP)-conjugated secondary antibodies (1:2500 dilution). After washing, bands were detected by ECL solution and visualized by x-ray films in dark room. Details of antibodies are: Anti-METTL3 (Aviva Systems Biology, Cat#: ARP39390_T100), Anti-SERPINA3 (Abcam, Cat#: ab129194), Anti-KAT2B (Abcam, Cat#: ab12188), Anti-KAT2A (Abcam, Cat#: ab217876), Anti-Phospho(Tyr) (Abcam, Cat#: ab179530), Anti-p-IKKα/β (Cell Signaling, Cat#: 2697), Anti-IKKγ (Cell Signaling, Cat#: 2685 Anti-IKKβ (Cell Signaling, Cat#: 8943), Anti-HA (Cell Signaling, Cat#: 3724), Anti-IκBα (Cell Signaling, Cat#: 4814), Anti-Phospho(Ser/Thr) (Cell Signaling, Cat#: 9631), Anti-Acetylated-Lysine (Cell Signaling, Cat#: 9441), Anti-SIRT1 (ProteinTech, Cat#: 13161-1-AP), Anti-SIRT2 (ProteinTech, Cat#: 19655-1-AP), Anti-SIRT6 (ProteinTech, Cat#: 13572-1-AP), Anti-SIRT7 (ProteinTech, Cat#: 12994-1-AP), Anti-His (ProteinTech, Cat#: 66005-1-Ig), Anti-Flag (ProteinTech, Cat#: 20543-1-AP), Anti-mouse IgG-HRP (Santa Cruz Biotechnology, Cat#: sc-516102), Anti-rabbit IgG-HRP (Santa Cruz Biotechnology, Cat#: sc-2357), Anti-Spi2a (Santa Cruz Biotechnology, Cat#: sc-57013), Anti-IKKα/β (Santa Cruz Biotechnology, Cat#: sc-7607), Anti-β-actin (Santa Cruz Biotechnology, Cat#: sc-47778), Anti-METTL14 (Sigma-Aldrich, Cat#: HPA038002), Anti-EIF3A (Cell Signaling, Cat#: 2538). Dilution for all of the primary antibodies is 1:1000.

### Macrophage depletion and reconstitution

Clodronate-containing liposomes were used to eliminate macrophages in terms of published investigations[46]. Briefly, mice were injected with one dose of 5 mg/ml clodronate-liposomes (0.2 ml/mouse, Encapsula NanoSciences, Cat#: CLD-8901) intravenously. After 48 hrs, depletion of macrophages was confirmed by MACS. Two days after this depletion treatment, mice were reconstituted by intravenous injection with $2 \times 10^6$ mouse-derived differentiated BMDMs which were dissolved in PBS. In other experiments, mice were reconstituted with lentivirus-transduced BMDMs. After 36 h, these reconstituted mice were intraperitoneally injected with PBS (pH7.0) or LPS (20 mg/kg) and monitored closely for up to 96 h.

### m$^6$A quantitation

The RNA m$^6$A Methylation kit (Epigentek, Cat#: P-9005-48) was applied to quantify the m$^6$A amounts in mRNAs or total RNAs according to the manufacturer's instructions. In brief, RNAs were mixed with m$^6$A capture antibodies in detection wells and incubated for 90 min at 37 °C. After a series of washes, the absorbance of each sample was monitored by a microplate reader at 450 nm. The isolation of mRNAs was performed by using the Dynabeads™ mRNA Purification Kit (ThermoFisher, Cat#: 61006) according to the manufacturer's protocol.

### m$^6$A RIP-qPCR

Total RNAs were isolated from BMDMs with or without 100 ng/ml LPS treatment by TRIzol Reagent, followed by enrichment of messenger RNAs using a Dynabeads mRNA Purification Kit. Messenger RNAs were fragmented by RNA Fragmentation Reagents (ThermoFisher, Cat#:

AM8740) and 4% of them served as input for internal control. Fragmented mRNAs were mixed with anti-m⁶A antibodies (Synaptic Systems, 1:100 dilution, Cat#: 202003) and incubated for 2 h in a cold room, followed by the purification of antibody-mRNA mixtures using the EpiMark $N^6$-Methyladenosine Enrichment Kit (New England Biolabs, Cat#: E1610S). Hexamer random primer was used to convert the IP-precipitated mRNA fragments to cDNAs, which were then quantified by real-time PCR. The enrichment of m⁶A was normalized to input controls. Related primers were listed in Supplementary Table 1. RIP-sequencing data were visualized by IGV 2.8.0.

#### Crosslinking-immunoprecipitation (CLIP) assay

CLIP assays were carried out according to previous studies with minor modifications[43]. In brief, BMDMs were treated with 100 mM 4-thiouridine for 14 h prior to 6-h LPS (100 ng/ml) treatment. After cold PBS washes, macrophages were irradiated with 0.15 J/cm² UV light. Then macrophages were collected and precipitated in cold PBS, followed by re-suspension in lysis buffer (150 mM KCl, 50 mM HEPES-KOH (pH 7.5), 1 mM NaF, 2 mM EDTA-NaOH (PH 8.0), 0.5 mM DTT, 0.5% NP-40, 1 μl/ml RNase inhibitor, and 1× protease inhibitor cocktail) with 10-min incubation on ice. The cell lysates were centrifuged at 13,000×$g$ and then the supernatants were filtered using a 0.2 μm filter prior to 15-min RNase T1 treatment (1 U/ml). 10% of lysates were saved as input, and the rest 90% of lysates were incubated with protein G magnetic antibody-conjugated beads for 1 h at cold room. The beads were enriched by a magnet and washed by IP wash buffer (0.05% NP-40, 50 mM HEPES-KOH, 0.5 mM DTT, 300 mM KCl, 1× protease inhibitor cocktail). The RNase T1 (100 U/ml) was used to treat beads for 15 min at 22 °C, prior to high-salt buffer (0.5 mM DTT, 50 mM HEPES-KOH (pH 7.5), 0.05% NP-40, 500 mM KCl and 0.5 ml/ml RNase inhibitor) washes. The beads were then re-dissolved in a proteinase K buffer (1.2 mg/ml proteinase K, 100 mM Tris-HCl (pH 7.4), 12.5 mM EDTA, 150 mM NaCl, and 2% SDS) with a 30 min incubation at 55 °C. Both input and precipitated RNAs were recovered using TRIzol Reagent. Reverse transcription and real-time PCR were carried out for RNA quantification. Related primers are listed in Supplementary Table 1. Details of antibodies are: anti-YTHDF1 (ProteinTech, Cat#: 17479-1-AP), anti-YTHDF2 (ProteinTech, Cat#: 24744-1-AP), anti-YTHDC1 (ProteinTech, Cat#: 14392-1-AP), anti-IGF2BP3 (ProteinTech, Cat#: 14642-1-AP), anti-FTO (Abcam, Cat#: ab92821), anti-IGF2BP1 (Cell Signaling, Cat#: 8482 S), anti-IGF2BP2 (Cell Signaling, Cat#: 14672 S), anti-WTAP (ProteinTech, Cat#: 60188-1-Ig), anti-ALKBH5 (ProteinTech, Cat#: 16837-1-AP), anti-YTHDF3 (Santa Cruz Biotechnology, Cat#: sc-377119), anti-YTHDC2 (Santa Cruz Biotechnology, Cat#: sc-249370). Dilution for all of the antibodies is 1:100.

#### mRNA decay assay

mRNA decay assays were carried out according to previous studies[43]. Briefly, BMDMs were challenged with 100 ng/ml LPS for 8 h, followed by the replacement of fresh media including actinomycin D (5 mg/ml). Total RNAs were collected at different time points (0, 2, 4, and 8 h) after actinomycin D treatment, and RT-real-time PCR was performed to quantify the mRNA transcripts.

#### Protein decay assay

Macrophages pre-treated by PBS or LPS (100 ng/ml) for 8 h were incubated with a medium including Cycloheximide (CHX, 50 μg/ml) for different hours as indicated. Cells were dissolved in RAPI buffer and then subjected to western blot analysis. The densitometric quantitation of bands was performed by ImageJ software.

#### NF-κB activity assay

NF-κB activity assay was performed using NF-κB luciferase reporter kit (BPS Bioscience, Cat#: 60614) according to the manufacturer's instruction. Macrophages with or without various treatments were co-transfected with NF-κB luciferase reporter and pRL-TK renilla luciferase reporter by jetPEI-Macrophage DNA Transfection Reagent (Polyplus-transfection, Cat#: 101000043). The luciferase activity was measured using a Dual Luciferase Assay System (Berthold Technologies).

#### Cell viability and death assays

Cell viability and death were detected using the CellTiter-Glo 2.0 kit (Promega, Catalog # G9241) and CytoTox 96 Non-Radioactive Cytotoxicity Assay kit (Promega, Catalog #G1780) according to the manufacturers' instructions, respectively.

#### CRISPR/Cas9-regulated knockout of *Ythdf1*, Spi*2a* and *METTL14*

sgRNA sequences targeting the *Ythdf1* gene (5'-AGCAGCCACTTCA ACCCCGC-3'), Spi*2a* gene (5'-AAATGTATCATTCGGGTCAA-3') and *METTL14* gene (5'-GCTGAAAGTGCCGACAGCAT-3') were subcloned into lentiCRISPRv2 vector (Addgene, catalog 52961) using the BsmBI restriction enzyme. Packaging plasmids (pMD2.G and psPAX2) and lenti-vector were co-transfected into HEK293T cells. 48 h after transfection, lentivirus was collected from the cell culture medium and added into macrophages with 4 μg/ml polybrene.

#### Cardiac function assessment

20 h After LPS treatment, mice were anesthetized, and cardiac functions such as heart rate (HR), left ventricle ejection fraction (LVEF), left ventricular wall thickness (LVWT) and left ventricular end-systolic volume (LVESV) were assessed by a vevo 770 ultrasound machine (Visual Sonics, Toronto, Canada).

#### F4/80-positive macrophage sorting

Fresh heart tissues were harvested and digested into single cells by a Papain Dissociation System (Worthington, Cat#: LK003150) based on the manufacturer's protocol. F4/80-positive macrophages were sorted using Anti-F4/80 MicroBeads UltraPure kit (Miltenyi Biotec, Cat #: 130-110-443) according to manufacturer's instructions. In brief, 10 μl of anti-F4/80 MicroBeads was mixed with the single-cell suspension and incubated in cold room for 15 min. After washing, macrophage-bead mixture was purified by magnetic separation columns and F4/80-positive macrophages were collected.

#### Newly translated protein assessment

A newly translated protein assessment was performed according to the previous investigation[43]. Briefly, BMDMs that were treated with or without 100 ng/ml LPS for 6 h were considered as 0 h samples. 3.5 or 4.5 h after treatment, the culture media of BMDMs were changed into methionine (Met)-free RPMI1640. 30 min later, the media were replaced with Met-free RPMI1640 including AHA (40 mM), and incubated for one hour (1 h samples) or two hours (2 h samples), respectively. After 6 h, BMDMs were lysed in 50 mM Tris-HCl (pH 8.0) with protease inhibitors and 1% SDS by sonication and incubated on ice for 30 min. Cell lysates were centrifuged at 12,000 × $g$ for 20 min and the supernatants were harvested. The Click & Go Protein Reaction Buffer Kit (Click Chemistry Tools, Cat#: 1262) and Biotin-PGE4-Alkyne were used to treat cell lysates for biotinylation. 10% of the recovered proteins were saved as input controls, and the other 90% of proteins were subjected to streptavidin-coated magnetic beads (High Capacity Streptavidin Magnetic Beads, Click Chemistry Tools, Cat#:1497-1) to enrich the AHA-labeled proteins. Both input and precipitated proteins were lysed in RIPA buffer, followed by western blot analysis.

#### Luciferase reporter assay

BMDMs were cultured in 24-well plates with differentiated medium and transduced with ctrl-, M14-WT, M14-Mut, M3-WT, M3-Mut or *Kat2b*-lentivirus for 48 h. After transduction, BMDMs were co-transfected with 500 ng pGL3-Promoter (control), pGL3-Spi*2a*-WT or

pGL3-Spi*2a*-Mut and pRL-TK renilla luciferase reporter using jetPEI-Macrophage DNA Transfection Reagent (Polyplus-transfection, Cat#: 101000043). 24 h after transfection, BMDMs were lysed and luciferase activities were measured by a Bio-Glo Luciferase Assay System kit (Promega, Cat#: G7941) according to the manufacturer's instruction. In another experiment, BMDMs were treated with LPS for 8 h after 24-h luciferase reporter vector transfection, and then luciferase activities were detected as described above.

### Lentiviral and plasmid constructs

Lentivirus overexpressing mouse IKKβ, METTL3, METTL14, KAT2A, KAT2B, SIRT1, SIRT2, SIRT6, SIRT7, Spi2a, FTO or DERPINA3F, and human IKKβ, METTL14, KAT2B, SOCS1 or SERPINA3 were generated via cloning the coding region of mouse *Ikkβ* [NM_001159774.1], *Mettl3* [NM_019721.2], *Mettl14* [NM_201638.2], *Kat2a* [NM_020004.5], *Kat2b* [NM_020005.4], *Sirt1* [NM_019812.3], *Sirt2* [NM_022432.4], *Sirt6* [NM_181586.4], *Sirt7* [NM_153056.3], Spi*2a* [NM_009251.2], *Fto* [NM_011936.2], *Serpina3f* [NM_001168294.1] cDNA or human *METTL14* [NM_020961.4], *SOCS1* [NM_003745.1], *IKKβ* [NM_001556.3], *KAT2B* [NM_003884.5], *SERPINA3* [NM_001085.5] cDNA into pLV[Exp]-Neo-EF1A lentiviral vector (VectorBuilder). Next, using these vectors as templates, an HA tag was added in the N-terminal of *Kat2b*, Spi*2a*, *SERPINA3*; a Flag tag was added in the N-terminal of *Kat2a*; a His tag was added in the N-terminal of *Mettl14*, *Ikkβ*, and *IKKβ*. Site-specific mutation of *METTL14* and *METTL3* was constructed by a QuickChange Site-Directed Mutagenesis Kit (Agilent) based on the manufacturer's protocols. Packaging plasmids and lenti-vector were co-transfected into HEK293T cells. 48 h after transfection, lentivirus was collected from the cell culture medium and added into macrophages with 4 μg/ml polybrenes. The fragment bearing m6A site in the cDNA of Spi*2a* was subcloned to the downstream of Luc gene of luciferase reporter vector pGL3-Promoter (Promega, Cat#: 200517) to generate the pGL3-Spi*2a* plasmid. pGL3-Spi*2a*-Mut was generated by mutating the m6A motif sequence 5'GA**A**CC3' to 5'GA**T**CC3' in pGL3-Spi*2a* plasmid using the Mutagenesis Kit above. All of the mutations were confirmed by DNA sequencing. Primers were listed in Supplementary Table 1.

### Cytokine measurement

Mouse or human TNF, IL-6, IL-1β, IFNγ, HMGB1, IL-1α and IL-33 concentrations in the cell culture media and serums were detected by ELISA kits according to the standard protocols. Data were collected by Gen5 (ver 3.10).

### Histology, immunohistochemical and TUNEL staining

Fresh hearts were harvested and fixed in 4% formaldehyde overnight at room temperature. The tissues were then processed, embedded and cut into sections (4 μm). Hematoxylin and eosin were used for histological examination. For immunohistochemical staining, the sections were boiled in sodium citrate (10 mM, pH 6.0) for 15 min prior to overnight anti-F4/80 antibody (ProteinTech, 1:100 dilution, Cat#: 28463-1-AP, RRID: AB_2881149) incubation in cold room. On day 2, sections were treated with anti-rabbit IgG-HRP-conjugated secondary antibodies (1:200 dilution), followed by 3,3'-diaminobenzidine visualization. TUNEL staining was performed using an In Situ Cell Death Detection Kit (Roche, Cat#: 12156792910) according to the manufacturer's instruction. Quantification of apoptosis was analyzed by counting the TUNEL-positive cells from 20 fields in each sample. The slides were observed under a microscope (Leica DM 2500). Quantitative analysis of images was performed by ImageJ (ver 1.4.3.67).

### Cecum ligation and puncture (CLP)

CLP procedure was carried out according to the published investigation[59]. Mice were treated with 100% oxygen containing 2.5%

isoflurane in the biological safety cabinet for anesthesia. Buprenorphine was administered at the dose of 0.05 mg/kg subcutaneously before skin sterilization. After making a midline abdominal incision, the cecum was exposed, ligated and perforated using a 20-gauge needle. A small amount of stool was exposed by squeezing cecum. After returning the cecum and closing the incision, warmed saline (0.5 ml) was subcutaneously administered. The sham procedure without cecal ligation and puncture was performed for control mice. Serums and macrophages were collected from mice at 20 h after surgery for further analyses.

### Lipopolysaccharide (LPS)-induced endotoxemia

Mice were intraperitoneally injected with PBS or 20 mg/kg LPS (Sigma-Aldrich, Cat#: L2630) and then closely monitored up to 96 hours. In another experiment, serums and peritoneal macrophages were collected from mice at 20 h after LPS injection for further analyses.

### LPS measurement

Serum LPS of human sera was detected using a commercial LPS Elisa kit according to the manufacture's protocol. The OD values were monitored by a microplate reader under 450 nm.

### RNA affinity chromatography

RNA affinity chromatography experiment was conducted in terms of previous investigations[19]. Biotin-labeled ssRNA oligonucleotides were from GE Dharmacon (Lafayette, CO). 0.4 pmol RNA oligonucleotide was mixed with 50 μl streptavidin magnetic beads in a binding buffer at 4 °C for 4 h. Next, the RNA bait-beads complexes were incubated with 200 μg extract from BMDMs in a cold room for 12 h. After washes, RNA–protein mixtures were dissolved in Laemmle buffer and subjected to western blot analysis.

### Statistical analysis

Data values were presented as means ± SD. The unpaired two-tailed Student's *t* test was applied for two group's comparisons, and multiple comparisons were carried out using one-way or two-way ANOVA. Comparisons related to animal survival rates were performed by the log-rank test. *P* values <0.05 were considered statistically significant. Statistical analysis was performed by GraphPad Prism v8.

### Reporting summary

Further information on research design is available in the Nature Portfolio Reporting Summary linked to this article.

## Data availability

All data files related to this paper are available in the Github repository (https://github.com/DrWangL/SepsisM6A20230201). Source data are provided with this paper.

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

## Acknowledgements

This study was supported by Stomatological Hospital of Shanxi Medical University grant RC2021-06 (J.D.), Shanxi Medical Key Technologies Program 2020XM08 (X.Y.W.), Shanxi Scholarship Council of China grant 2021-087 (X.Y.W.), Hainan Provincial Natural Science Foundation of China grant 820MS131 (W.L.), Hainan Provincial Key Research and Development Program ZDYF2020147 (Y.D.), Hainan Natural Science Foundation Innovation Research Team Project 2018CXTD350 (Y.D.), National Natural Science Foundation of China grants 82160074 (W.L.), 81800499 (J.D.), 81960565 (Y.D.), 81560275 (Y.D.), Youth Science and Technology Project of Hainan General Hospital QN202005 (W.L.), Hainan Province Clinical Medical Center (Y.D.), Excellent Talent Team of Hainan Province QRCBT202121 (Y.D.) and Hainan Province Clinical Medical Center (W.L.).

## Author contributions

J.D. conceived and designed the research. X.W., Y.D., R.L., R.Z., and X.G. performed the experiments. R.G., M.W., and Y.H. provided assistant work for experiments. R.Z., W.L., F.Z., and B.Z. collected human samples. W.L. analyzed data. W.L. supervised the patient research. J.D. wrote the manuscript. J.D., X.W., Y.D., R.L., and W.L. acquired funding. All authors read and approved the final manuscript.

## Competing interests

The authors declare no competing interests.
