## [Peer Review File · Nature Communications]

REVIEWER COMMENTS

Reviewer #1 (expertise in m6A RNA modification, RNA biology):

In this article, the authors seek to define the role of m6A methylation of Spi2A in the attenuation of macrophage production of cytokines during sepsis.

The claim is that LPS exposure causes acetylation of METTL14 to increase its stability, and thereby promoting m6A methylation in macrophages. One of the targets during sepsis is Spi2A, and the m6A methylation increase of Spi2A leads to greater translation. There are then some studies which show the effect of Spi2A on inflammatory signals/cytokines. The authors appear to identify the potential reader protein of this epimark in this context (YTHDF1), and provide evidence that this increases EIF3A binding to Spi2A to increase translation.

Broadly, I thought this paper was quite well argued and often convincing. The mechanism of m6A affecting translation is fairly well understood at this point, so I would expect the involvement of this in sepsis/inflammatory response to be more impactful to those fields, than to the epitranscriptome/mRNA modifications field.

There are some issues that need to be addressed before publication in my opinion. These include some major and minor aspects:

Major:

The materials and methods should not be in the supplementary data, this should be in the main text.

It would be possible, with some difficulty to replicate based on the information provided.

In figure 2a, m6A quantification is performed on total RNA. This gives little insight into the m6A content on messenger RNA, given how much more m6A exists in other RNA species (which constitute the majority of the transcriptome). It should at least be performed on poly(A) enriched or rRNA depleted samples.

Fig 2e. Though the western blots to determine METTL14 post-translation modifications are convincing, there does not appear to be any confirmation that the immunoprecipitation has been effective. The authors should show a blot for another protein in the IP Anti-METTL14 to confirm purity (or show a SDS-PAGE demonstrating single band purity).

I am not convinced the authors have shown that YTHDF1 readers Spi2A and recruits eIF3A. With a cloned 3'UTR containing the modified site, a gel shift assay (EMSA) with recombinant YTHDF1 would confirm interaction, and be able to show if this is dependent on m6A methylation.

Fig 3a appears to show increases in m6A sites across the whole transcript of Spi2A, but the description of this experiment is not very clear. What is the "input" shown? This is important, as given it is uncommon (but not unheard of) to find m6A outside of the 3'UTR/near stop codon, this increase in m6A enriched peaks could just represent some increased expression caused by LPS treatment, or be reflecting the changes to Spi2A half life observed in fig 3d & e - and the full length being enriched through a function of technical error combined with enriched transcript. The qPCR following this up is somewhat more convincing, but to my mind wouldn't determine if the result in fig 3a is from an increased Spi2A amount, independent of m6A. From Fig 3a it seems like Spi2A may be one of the less common transcripts that is m6A-marked more broadly.

On the mechanism - YTHDF1 generally increases. This will have wide-reaching effects on translation. The proposed mechanism is that Spi2A is better translated when m6A modified, as it is "read" by YTHDF1. This is shown by AHA labelling and precipitation prior to western blot. This is quite convincing, but the authors need to show another precipitated target not altering in the same way. I.e. Is YTHDF1 altering global translation? Currently, the presented data do not rule this out. At this point, there are a range of possible proteins which could be binding Spi2A mRNA and affecting its stability - is this the same YTHDF1 binding to affect translation - or are the translation effects caused by greater presence of Spi2A mRNA resulting from increased half life/greater expression?

Minor/Formatting/Style:

It is often difficult to piece together what is occurring from the figures and methods, as the required information is often split between the two.

Ln 80/82 - There are now many more modifications identified on RNA, the modomics website would be a better reference for this than the chosen one.

Reviewer #2 (expertise in inflammation in sepsis):

In this detailed and apparently well-conducted study, the authors report that LPS upregulates serine protease inhibitor 2A (SPI2A) in BMDM. This appears to be mediated by the enhanced m6A methylation. Loss and gain function experiments were also conducted to confirm the importance of SPI2A and m6A methylation. Human macrophage data, heart injury and sepsis patient information were also provided. They conclude that m6A methylation of SPI2A is a novel factor responsible for a negative feedback control of activated macrophages in sepsis.

1. Despite statistical significance, many in vitro experiments were conducted at n=3. An increase of sample size is needed.
2. Except for Figure S5c-j in which CLP was used, LPS or endotoxemia does not precisely represent clinical sepsis. A large number of DAMPs in sepsis are equally important.
3. There is no information showing the effect of LPS on SPI2A and m6A methylation is mediated by TLR4. Can they eliminate endotoxin's effects by block TLR4 in macrophages?
4. What is the translational approach of this finding? How can one modulate SPI2A and m6A methylation?
5. What are the effects on other organs such as the lungs and liver?
6. The survival studies were conducted for about 4 days. What happened with the animals at 10 days?
7. It would be interesting if peritoneal and tissue macrophages can be isolated after sepsis (not endotoxemia) to determine the changes of SPI2A and m6A methylation.

Reviewer #3 (expertise in Spi2A, erythropoiesis):

This study builds from the observation of LPS-induced increases in Spi2A transcripts & protein, through to a model in which increased KAT2B activity boosts METTL14, which leads to m6A acetylation of Spi2A mRNA. This results in stabilization, heightened Spi2A and levels. Spi2A then interacts with IKKB to limit the NFK-B pathway to inflammatory cytokine expression in macrophage. This pathway is dissected and tested in mouse cells and models, and also put to several tests in human macrophage. This includes correlates of Spi2A and inflammation and disease severity in PB and monocytes from sepsis patients.

Critiques of specific experiments and interpretations follow, including summations.

Abstract: Clearly written, including a concise recap of the prime components of a 5-step mechanism for Spi2A attenuating NFKB- regulated inflammatory cytokine regression. This includes in vivo assessments of expressed, and inhibited M6A methylation on myocardial damage. One summary point that might be more clearly defined is the "disruption" by Spi2A of IKK complexes (vs hinders complex formation?)

Introduction: Quite informative for this complex pathway and its components. One added component that might be introduced is similarity (vs differences) in mouse vs human Spi2A (including specificities as serpins) vs other activities.

Data, Figures and Interpretations

Fig-1: Data are solid that reveal Spi2A induction by LPS, and IKKB expression in BMDM's. Bay 11-7082 dosing results are nicely intuited, including subsequent observations of LPS in Spi2A mRNA decay. (In 2i, changes in Spi2A stability are ~2-fold).

Fig-2: LPS is shown to significantly increase METTL14 levels in BMDM's (2a-d), including its acetylation (2e). KAT2B further is implicated as the acetylase, and co-IP'd partner (2g-2k). For LM-dihydride, references or data on specificity would be helpful.

Among nuclear SIRT5, SIRT1 is implicated as a deacetylase for METTL14, and this is demonstrated via SIRT1 expression in BMDM's. For METTL14, K, R mutation mapping points to K398 as an acetylation site (of KAT2B) that can stabilize METTL14. And mutation to K398R limited 1/2-life protection. This builds an interesting case for LPS to KAT2B to METTL14 stabilization.

Fig-3: Using M148/8 vs -/- BMDM, evidence next is provided for M14 mediated Spi2A transcript methylation, and M14 deletion on E1F3A binding. Spi2A reporter assays added credence for M14's actions including Spi2A and M14 mutant controls.

Via CRISPR/C9 YTHDF1 and FTO are implicated as readers of m6A Spi2A demethylation. This is informative (but here connections to LPS signaling are a bit less clear, but aided by Fig 3q (Q: are p-values for FTO [+] vs [-] LPS?)).

Fig-4 focuses on Spi2A and demonstrates GOF and LOF decreases, and increases in LPS induced cytokines in macrophage. (Note: "Spi2A" mouse and "SPI2A" for human would be helpful). This extends to Spi2A effects on NFkB activity in BMDM (4e). (Effects here, however, are <2-fold). 4f provides evidence for Spi2A action on limiting IKBa. Spi2A co-IPs with IKKB kinase (kinase inhibitor of inhibitory IKK complex). Does Spi2A affect IKKB activity?

Fig-5 transitions to in vivo studies, using Spi2A K/O macrophage reconstitution, and LPS sepsis challenge.

5a-c show increased mortality and decreased cytokine levels. 5d-g and 5h,i clearly define heightened cardiac compromise, including output (s5a). Infiltrated macrophage levels are similar for Spi2A KO vs +/+ macrophage. Because Spi2A K/O can compromise cell viability under stress, it would be helpful (if not important) to know the representation of reconstituted Spi2A KO vs +/+ cells (e.g., by cell trace marking, or similar). CPL studies add a clinically relevant model to studies.

Fig-6: The focus on M14 is a good connecting piece, including the use of KO mice, including rescue reconstitutions, and phenotyping through to heart (dys)function. This extends to demethylase roles (YTHDK1, and FTO). One limitation is the likely broader effects of M14 and YTHDF1, FTO) in macrophage- but results do support this 5-level model to Spi2A and NFkB regulation. GOF studies for Spi2A further strengthen this model.

Fig-S10 helps to address the question of functional commonality between mouse Spi2A (Serpina3g) and human (SPI2A) (SERPINA3). In human macrophage, GOF h-SPI2A limited LPS-induced cytokines, and NFkB activation, and shared common AA's for effects on IKK. Here, it would be useful to test the ability of h-SPI2A to exert such effects in mouse macrophage (and ultimately mice).

Fig-6 (continued) ...studies in sepsis patients demonstrate correlations between SERPINA3 and m6A levels; and a negative correlation with cardiac dysfunction markers. This is a small patient set, but trends are consistent with SPI2A effects and action modes generalized in mouse models.

Overall, this work goes far to piece together integrated players in a newly proposed Spi2A signaling circuit that blunts LPS-induced inflammation, and cardiovascular damage.

Responses to reviewers' comments

Reviewer #1 (expertise in m6A RNA modification, RNA biology):

In this article, the authors seek to define the role of m6A methylation of Spi2A in the attenuation of macrophage production of cytokines during sepsis. The claim is that LPS exposure causes acetylation of METTL14 to increase its stability, and thereby promoting m6A methylation in macrophages. One of the targets during sepsis is Spi2A, and the m6A methylation increase of Spi2A leads to greater translation. There are then some studies which show the effect of Spi2A on inflammatory signals/cytokines. The authors appear to identify the potential reader protein of this epimark in this context (YTHDF1), and provide evidence that this increases EIF3A binding to Spi2A to increase translation. Broadly, I thought this paper was quite well argued and often convincing. The mechanism of m6A affecting translation is fairly well understood at this point, so I would expect the involvement of this in sepsis/inflammatory response to be more impactful to those fields, than to the epitranscriptome/mRNA modifications field.

Response: We thank Reviewer #1 for the constructive comments. The manuscript has been rewritten and we have carefully addressed these comments. The involvement of m⁶A and Spi2A in sepsis/inflammatory response has been well explained in this study. We explored the roles of m⁶A and Spi2A in two sepsis animal models and human samples. Since there are a lot of data in the main text and these responses will be also published online, we prefer to show some data here.

There are some issues that need to be addressed before publication in my opinion. These include some major and minor aspects:

Major:

The materials and methods should not be in the supplementary data, this should be in the main text. It would be possible, with some difficulty to replicate based on the information provided.

Response: Thank you! The materials and methods part has been moved to the main text.

In figure 2a, m6A quantification is performed on total RNA. This gives little insight into the m6A content on messenger RNA, given how much more m6A exists in other RNA species (which constitute the majority of the transcriptome). It should at least be performed on poly(A) enriched or rRNA depleted samples.

Response: The new experiments on mRNAs have been conducted in this

revision.

Fig 2e. Though the western blots to determine METTL14 post-translation modifications are convincing, there does not appear to be any confirmation that the immunoprecipitation has been effective. The authors should show a blot for another protein in the IP Anti-METTL14 to confirm purity (or show a SDS-PAGE demonstrating single band purity).

I am not convinced the authors have shown that YTHDF1 readers Spi2A and recruits eIF3A. With a cloned 3'UTR containing the modified site, a gel shift assay (EMSA) with recombinant YTHDF1 would confirm interaction, and be able to show if this is dependent on m⁶A methylation.

Response: These are good suggestions. In figure 2e, β -actin has been selected as another protein to confirm purity. To confirm interaction, we synthesized methylated single-stranded RNA bait (ss-m⁶A, with the consensus sequence GA(m⁶A)CC) or unmethylated control RNA (ss-A) in the 3'UTR of *Spi2A* (Figure 3i) for RNA pull-down in BMDMs according to published paper (PMID: 29476152). As shown in figure 3i, biotin-labelled RNA oligonucleotides containing m⁶A, but not control RNA, is able to bind YTHDF1 and recruit EIF3A in BMDMs. Details of method can be found in the published paper (PMID: 29476152).

Fig 3a appears to show increases in m⁶A sites across the whole transcript of *Spi2A*, but the description of this experiment is not very clear. What is the "input" shown? This is important, as given it is uncommon (but not unheard of) to find m⁶A outside of the 3'UTR/near stop codon, this increase in m⁶A enriched peaks could just represent some increased expression caused by LPS treatment, or be reflecting the changes to *Spi2A* half life observed in fig 3d & e - and the full length being enriched through a function of technical error combined with enriched transcript. The qPCR following this up is somewhat more convincing, but to my mind wouldn't determine if the result in fig 3a is from an increased *Spi2A* amount, independent of m⁶A. from Fig 3a it seems like *Spi2A* may be one of the less common transcripts that is m⁶A-marked more broadly.

Response: These are great comments. More description of this experiment has been shown in the figure legend. To provide more convincing data, we designed primers around the other two peaks in the coding region (site 2 and 3) as shown in panel a, and then performed m⁶A-RIP-qPCR (panel b) and CLIP (panel c). Results in panel b show that there are m⁶A modifications in the two sites outside of the 3'UTR/near stop codon. CLIP data in panel c confirm the interaction between METTL14/METTL3 and m⁶A in site 2 and 3. Together, these data claim that the increased *Spi2A* amount is dependent on m⁶A modification. We hope this reviewer will agree with us.

Figure legend.

a. Read density of *Spi2A* transcript in M14^{fl/fl} and M14^{-/-} BMDMs with PBS or LPS treatment.

b. RIP-qPCR assays verifying the m⁶A peaks of *Spi2A* transcript of m⁶A-IP sequencing data.

c. CLIP assays demonstrating the binding of METTL14 or METTL3 to the *Spi2A* m⁶A site in BMDMs after LPS treatment.

** $P < 0.01$, *** $P < 0.001$, **** $P < 0.0001$ versus PBS control group; # $P < 0.05$ versus corresponding LPS group. Data depict mean \pm SD.

On the mechanism - YTHDF1 generally increases. This will have wide-reaching effects on translation. The proposed mechanism is that SPI2a is better translated when m⁶A modified, as it is “read” by YTHDF1. This is shown by AHA labelling and precipitation prior to western blot. This is quite convincing, but the authors need to show another precipitated target not altering in the same way. I.e. Is YTHDF1 altering global translation? Currently, the presented data do not rule this out. At this point, there are a range of possible proteins which could be binding *Spi2A* mRNA and affecting its stability - is this the same YTHDF1 binding to affect translation - or are the translation effects caused by greater

presence of *Spi2A* mRNA resulting from increased half life/greater expression?

Response: These are good questions.

We detected β -actin expression in the IP samples and found that β -actin levels are not affected by YTHDF1 (Figure 3j).

Adenylate-uridylate-rich elements (AU-rich elements; AREs) are found in the 3' untranslated region (UTR) of many messenger RNAs (mRNAs). AREs are one of the most common determinants of RNA stability in mammalian cells (PMID: 8578590). HUD (also called ELAVL4) binds to AREs and increases the half-life of ARE-bearing mRNAs (PMID: 17853436; PMID: 11948657). Here, we showed the AU-rich element in the mRNA of *Spi2A* (panel a) and HUD overexpression increases *Spi2A* mRNA stability in BMDMs (panel b). However, HUD has no physical interaction with EIF3A/EIF3B (panel c) and overexpression of HUD in BMDMs does not increase the recruitment of EIF3A/EIF3B to *Spi2A* mRNA (panel d). These data demonstrate that HUD can increase *Spi2A* mRNA stability, but have no effect on *Spi2A* mRNA's translation efficiency. Thus, based on our data, different *Spi2A* mRNA-binding proteins have distinct roles in regulating mRNA stability and translation efficiency.

Figure legend.

a. Schematic illustration of AU-rich element in the mRNA of *Spi2A*.

b. RNA decay assays of *Spi2A* mRNA in BMDMs with or without HUD overexpression.

c. BMDMs were transduced with control-or HUD-lentivirus and immunoprecipitated with HUD antibody, followed by western blot analyses.

d. CLIP assays demonstrating the binding of EIF3A or EIF3B to the *Spi2A* AU-rich element in BMDMs with or without HUD overexpression.

** $P < 0.01$ versus control group. Data depict mean \pm SD.

Minor/Formatting/Style:

It is often difficult to piece together what is occurring from the figures and methods, as the required information is often split between the two.

Ln 80/82 - There are now many more modifications identified on RNA, the modomics website would be a better reference for this than the chosen one.

Response: This manuscript has been rewritten according to your suggestions.

Reviewer #2 (expertise in inflammation in sepsis):

In this detailed and apparently well-conducted study, the authors report that LPS upregulates serine protease inhibitor 2A (SPI2A) in BMDM. This appears to be mediated by the enhanced m6A methylation. Loss and gain function experiments were also conducted to confirm the importance of SPI2A and m6A methylation. Human macrophage data, heart injury and sepsis patient information were also provided. They conclude that m6A methylation of SPI2A is a novel factor responsible for a negative feedback control of activated macrophages in sepsis.

Response: We thank Reviewer #2 for the constructive comments. We provided more data according to your good suggestions. Since there are a lot of data in the main text and these responses will be also published online, we prefer to show some data here.

1. Despite statistical significance, many in vitro experiments were conducted at n=3. An increase of sample size is needed.

Response: Thank you! Except for figure 2o and supplementary figure 2g, the biological replicates for in vitro experiments are 5 in this revision. For the results of figure 2o and supplementary figure 2g, we used several approaches such as active and inactive mutations to verify the same finding.

2. Except for Figure S5c-j in which CLP was used, LPS or endotoxemia does not precisely represents clinical sepsis. A large number of DAMPs in sepsis are equally important.

Response: These are really good suggestions. CLP model was involved in all of the animal studies in this revision (supplementary figure 5, supplementary figure 6o-z, supplementary figure 7h-m, supplementary figure 8o-z and supplementary figure 9f-q). Serum DAMPs such as HMGB1, IL-1 α and IL-33 were also detected (Figure 5c and 5l, supplementary figure 5e, supplementary figure 6c,6j,6q and 6w, supplementary figure 7c and 7j, supplementary figure 8c,8j,8q and 8w, supplementary figure 9c,9h and 9n).

3. There is no information showing the effect of LPS on SPI2A and m6A methylation is mediated by TLR4. Can they eliminate endotoxin's effects by block TLR4 in macrophages?

Response: To answer this question, we deleted *Tlr4* gene in BMDMs and found that the deletion of *Tlr4* is able to block m⁶A and SPI2A increases induced by LPS (panel a-c). These data demonstrate that endotoxin's effects on m⁶A methylation and *Spi2A* are regulated by TLR4.

Figure legend.

a. Western blot showing TLR4 and SPI2A levels in control or TLR4-knockout BMDMs with or without LPS treatment.

b. Measurements of m⁶A levels in control or TLR4-knockout BMDMs with or without LPS treatment.

c. qPCR showing *Spi2A* mRNA levels in control or TLR4-knockout BMDMs with or without LPS treatment.

*** $P < 0.001$, **** $P < 0.0001$ versus PBS control group; ## $P < 0.01$ versus corresponding LPS group. Data depict mean \pm SD.

4. What is the translational approach of this finding? How can one modulate SPI2A and m⁶A methylation?

Response: This is an interesting question. In this study, we found that m⁶A methylation-regulated SPI2A is important for suppressing inflammatory responses. Furthermore, SIRT1 is able to accelerate METTL14 protein degradation and decrease m⁶A levels, which leads to the inhibition of SPI2A expression. Therefore, SIRT1 inhibitor might be a novel approach for sepsis management in clinic.

5. What are the effects on other organs such as the lungs and liver?

Response: To answer this question, we performed HE staining and qPCR to detect lungs and liver injuries. As shown in panel a here, *Spi2A* deletion in macrophages worsens lung injuries in the CLP animal model, while forced expression of *Spi2A* in macrophages ameliorates lung injuries. We next measured cytokines levels in lung and liver tissues in both LPS and CLP animal models using qPCR. As shown in panel b, the loss of *Spi2A* or METTL14 exacerbates inflammatory responses in lung and liver tissues, while overexpression of *Spi2A* suppresses cytokine productions. Together, *Spi2A* and m⁶A methylation play protective roles in lung and liver injuries under sepsis condition.

Figure legend:

Macrophage-depleted wild type mice were reconstituted with wild type BMDMs transduced with sgCtrl- or sg*Spi2A*-lentivirus prior to sham or CLP surgery, then the HE staining was performed.

Figure legend:

Quantitative PCR analyses of cytokine expression in lung and liver tissues from mice with macrophage-specific *Spi2A/Mettl14* deletion or *Spi2A* overexpression in sepsis models. **

$P < 0.01$, *** $P < 0.001$ versus corresponding control group. Data depict mean \pm SD.

6. The survival studies were conducted for about 4 days. What happened with the animals at 10 days?

Response: This is a great question. Macrophages are differentiated from bone marrow stem cells and precursor cells within tissues. In the macrophage depletion and reconstitution system, timing is crucial in the experimental design in order to generate meaningful results. Following clodronate-liposome-mediated macrophage depletion, the endogenous macrophages can repopulate the mouse within 1 week (PMID: 34917981). As such, the entire experiment (including macrophages depletion and reconstitution) has to be completed within 7 days. We hope this reviewer will agree with us.

7. It would be interesting if peritoneal and tissue macrophages can be isolated after sepsis (not endotoxemia) to determine the changes of SPI2A and m6A methylation.

Response: As shown in panel a here, both m⁶A and Spi2A levels are increased in peritoneal macrophages derived from mice in CLP models, while depletion of METTL14/METTL3 in macrophages inhibits m⁶A and Spi2A increases in the CLP animal models. Since there is no F4/80+ macrophages in the healthy heart tissues, we only isolated heart macrophages from mice with CLP surgery. As shown in panel b, m⁶A and Spi2A levels in heart macrophages are also increased under septic conditions.

Figure legend

a. m⁶A levels detected by ELISA and Spi2A levels detected by qPCR in peritoneal

macrophages from mice with sham or CLP surgery.

b. m⁶A levels detected by ELISA and Spi2A levels detected by qPCR in heart macrophages from mice with CLP surgery.

*** $P < 0.001$, **** $P < 0.0001$ versus M14^{ff} or M3^{ff} sham control group; ## $P < 0.01$, ### $P < 0.001$ versus corresponding CLP group. Data depict mean \pm SD.

Reviewer #3 (expertise in Spi2A, erythropoiesis):

This study builds from the observation of LPS-induced increases in Spi2A transcripts & protein, through to a model in which increased KAT2B activity boosts METTL14, which leads to m6A acetylation of Spi2A mRNA. This results in stabilization, heightened Spi2A and levels. Spi2A then interacts with IKKB to limit the NFK-B pathway to inflammatory cytokine expression in macrophage. This pathway is dissected and tested in mouse cells and models, and also put to several tests in human macrophage. This includes correlates of Spi2A and inflammation and disease severity in PB and monocytes from sepsis patients.

Response: We thank Reviewer #3 for the constructive comments. We have addressed all of the questions you proposed. Since there are a lot of data in the main text and these responses will be also published online, we prefer to show some data here.

Critiques of specific experiments and interpretations follow, including summations.

Abstract: Clearly written, including a concise recap of the prime components of a 5-step mechanism for Spi2A attenuating NFKB- regulated inflammatory cytokine regression. This includes in vivo assessments of expressed, and inhibited M6A methylation on myocardial damage. One summary point that might be more clearly defined is the “disruption” by Spi2A of IKK complexes (vs hinders complex formation?)

Response: Thank you. This sentence has been corrected according to your suggestion.

Introduction: Quite informative for this complex pathway and its components. One added component that might be introduced is similarity (vs differences) in mouse vs human Spi2A (including specificities as serpins) vs other activities.

Response: More information on mouse and human Spi2A have been described in this revision.

Data, Figures and Interpretations

Fig-1: Data are solid that reveal Spi2A induction by LPS, and IKKB expression in BMDM's. Bay 11-7082 dosing results are nicely intuited, including subsequent observations of LPS in Spi2A mRNA decay. (In 2i, changes in Spi2A stability are ~2-fold).

Response: Thank you for your comments.

Fig-2: LPS is shown to significantly increase METTL14 levels in BMDM's (2a-d), including its acetylation (2e). KAT2B further is implicated as the acetylase, and co-IP'd partner (2g-2k). For LM-dihydride, references or data on specificity would be helpful.

Response: Thank you for your comments. The reference for LM-dihydride is: Moustakim M, et al. Discovery of a PCAF Bromodomain Chemical Probe. *Angew Chem Int Ed Engl.* 2017 Jan 16;56(3):827-831. (PMID: 27966810)

Among nuclear SIRT5, SIRT1 is implicated as a deacetylase for METTL14, and this is demonstrated via SIRT1 expression in BMDM's. For METTL14, K, R mutation mapping points to K398 as an acetylation site (of KAT2B) that can stabilize METTL14. And mutation to K398R limited 1/2-life protection. This builds an interesting case for LPS to KAT2B to METTL14 stabilization.

Response: Thank you for your comments.

Fig-3: Using M148/8 vs -/- BMDM, evidence next is provided for M14 mediated Spi2A transcript methylation, and M14 deletion on E1F3A binding. Spi2A reporter assays added credence for M14's actions including Spi2A and M14 mutant controls.

Response: Thank you for your comments.

Via CRISPR/C9 YTHDF1 and FTO are implicated as readers of M6A Spi2A demethylation. This is informative (but here connections to LPS signaling are a bit less clear, but aided by Fig 3q (Q: are p-values for FTO [+] vs [-] LPS?).

Response: Thank you for your comments. For the statistical analysis, ** P < 0.01, *** P < 0.001, **** P < 0.0001 versus PBS control group; ## P < 0.01 versus corresponding LPS group.

Fig-4 focuses on Spi2A and demonstrates GOF and LOF decreases, and increases in LPS induced cytokines in macrophage. (Note: "Spi2A" mouse and "SPI2A" for human would be helpful). This extends to Spi2A effects on NFkB activity in BMDM (4e). (Effects here, however, are <2-fold). 4f provides evidence for Spi2A action on limiting IKBa. Spi2A co-IPs with IKKB kinase (kinase inhibitor of inhibitory IKK complex). Does Spi2A affect IKKB activity?

Response: Thank you. Spi2A is the format for gene and mRNA, but SPI2A is for protein. For 4e, although effects here are <2-fold, the statistical analyses are significant. To determine the roles of Spi2A in IKKβ activity, we overexpressed SPI2A and β-actin (control) in macrophages and used a commercial kit

(Promega, Catalog #: V4502) to measure IKK β activity. As shown here, Spi2A is able to suppress IKK β activity in a dose-dependent manner.

Fig-5 transitions to in vivo studies, using Spi2A K/O macrophage reconstitution, and LPS sepsis challenge.

5a-c show increased mortality and decreased cytokine levels. 5d-g and 5h,i clearly define heightened cardiac compromise, including output (s5a). Infiltrated macrophage levels are similar for Spi2A KO vs +/+ macrophage. Because Spi2A K/O can compromise cell viability under stress, it would be helpful (if not important) to know the representation of reconstituted Spi2A KO vs +/+ cells (e.g., by cell trace marking, or similar). CPL studies add a clinically relevant model to studies.

Response: This is a very good comment. We deleted *Spi2A* in both BMDMs and peritoneal macrophages, followed by PBS or LPS treatment. As shown in panel a, neither Spi2A deletion nor LPS challenge affects macrophage viability. To further confirm it in vivo, we eliminated macrophages in CD45.2 mice using Clodronate-containing liposomes and reconstituted these CD45.2 mice with control or Spi2A-KO CD45.1 BMDMs. 2 days after reconstitution, peritoneal macrophages were collected and sorted by MagniSort™ Mouse CD45.1 Positive Selection Kit (ThermoFisher, Catalog#: 8802-6848-74) and MagniSort™ Mouse CD45.2 Positive Selection Kit (ThermoFisher, Catalog#: 8802-6849-74). As shown in panel b, the cell number and cell viability of reconstituted Spi2A-KO and control CD45.1+ macrophages are comparable. We next deleted macrophages in CD45.1 mice and reconstituted with CD45.2 macrophages to measure cell number and cell viability conversely. As shown in panel c, the cell number and cell viability of reconstituted Spi2A-KO and control CD45.2+ macrophages are comparable.

Figure legend

- a. Cell viability of control and Spi2A-KO macrophages with or without LPS treatment.
- b. Endogenous macrophages in CD45.2 mice were deleted and these mice were reconstituted with control or Spi2A-KO CD45.1+ BMDMs. The reconstituted macrophages were collected from peritoneal cavity and sorted by CD45.1/CD45.2 Positive Selection Kits. Cell number of these macrophages was quantified and cell viability was measured.
- c. Endogenous macrophages in CD45.1 mice were deleted and these mice were reconstituted with control or Spi2A-KO CD45.2+ BMDMs. The reconstituted macrophages were collected from peritoneal cavity and sorted by CD45.1/CD45.2 Positive Selection Kits. Cell number of these macrophages was quantified and cell viability was measured.

Fig-6: The focus on M14 is a good connecting piece, including the use of KO mice, including rescue reconstitutions, and phenotyping through to heart (dys)function. This extends to demethylase roles (YTHDK1, and FTO). One limitation is the likely broader effects of M14 and YTHDF1, FTO) in macrophage- but results do support this 5-level model to Spi2A and NFKB regulation. GOF studies for Spi2A further strengthen this model.

Response: Thank you for your positive comments.

Fig-S10 helps to address the question of functional commonality between mouse Spi2A (Serpina3g) and human (SPI2A) (SERPINA3). In human macrophage, GOF h-SPI2A limited LPS- induced cytokines, and NFKB activation, and shared common AA's for effects on IKK. Here, it would be useful

to test the ability of h-SPI2A to exert such effects in mouse macrophage (and ultimately mice).

Response: In mouse BMDMs, overexpression of human SERPINA3 suppressed cytokine levels and NF- κ B activity upon LPS treatment (panel a-c). Moreover, overexpression of SERPINA3 in macrophages could ameliorate inflammatory responses and heart injury in CLP animal model (panel d-g).

Figure legend

a-c. Mouse BMDMs overexpressing human SERPINA3 were treated with or without LPS. Cytokine mRNA levels in BMDMs were detected by qPCR (a), NF- κ B activity in BMDMs were detected by luciferase reporter assay (b) and cytokine secretions in the culture media of BMDMs were detected by ELISA (c).

d-g. The endogenous macrophages in wild type mice were deleted and these mice were reconstituted with BMDMs overexpressing human SERPINA3, followed by sham or CLP surgery. (d) Serum cytokine levels were determined by ELISA, (e-f) cytokine levels in peritoneal macrophages (e) and heart macrophages (f) were measured by qPCR, (g) Dead cells in heart tissues were measured by TUNEL assays.

** $P < 0.01$, *** $P < 0.001$, **** $P < 0.0001$ versus control group; ## $P < 0.01$, ### $P < 0.001$ versus corresponding LPS or CLP group. Data depict mean \pm SD.

Fig-6 (continued) ...studies in sepsis patients demonstrate correlations between SERPINA3 and m6A levels; and a negative correlation with cardiac dysfunction markers. This is a small patient set, but trends are consistent with SPI2A effects and action modes generalized in mouse models.

Response: Thank you. Due to this pandemic, we can not recruit a large number of patients for this study. We hope this reviewer will agree with us.

REVIEWERS' COMMENTS

Reviewer #1 (expert in m6A RNA modification, RNA biology):

The authors have sought to define the role of m6A methylation of SPI2A in the attenuation of macrophage production of cytokines during sepsis, the previous iteration of the manuscript was good but in need of additional experiments, and this has made some improvements.

Much of the work done to address reviewer comments has helped the paper, and it is now a convincing story. However, the methods used to isolate and confirm the purity of the mRNA prior to m6A detection are still unclear from the text. Figure 2A now shows measurements of m6A/A on mRNA, but the methods describe a method for total RNA. This seems to be an important, early, point of the work - that LPS stimulation results in increased m6A levels in mRNA - so the authors need to describe how they isolated mRNA for reproducibility, and ideally show the level of purity (e.g. by Agilent bioanalyser or similar).

Beyond this, I feel the authors have address my comments well with additional experiments, text, and data.

Reviewer #2 (expert in inflammation in sepsis):

Thank you for adequately addressing my various concerns.

Reviewer #2 acting as mediator for Reviewer #3 (absent):

I have carefully reviewed authors' responses to Reviewer #3's comments and concerns. I believe that the authors have adequately addressed those issues and revised the manuscript accordingly. My recommendation: acceptable for publication.

Responses to reviewers' comments

Reviewer #1 (expertise in m6A RNA modification, RNA biology):

The authors have sought to define the role of m6A methylation of SPI2A in the attenuation of macrophage production of cytokines during sepsis, the previous iteration of the manuscript was good but in need of additional experiments, and this has made some improvements.

Much of the work done to address reviewer comments has helped the paper, and it is now a convincing story. However, the methods used to isolate and confirm the purity of the mRNA prior to m6A detection are still unclear from the text. Figure 2A now shows measurements of m6A/A on mRNA, but the methods describe a method for total RNA. This seems to be an important, early, point of the work - that LPS stimulation results in increased m6A levels in mRNA - so the authors need to describe how they isolated mRNA for reproducibility, and ideally show the level of purity (e.g. by Agilent bioanalyser or similar).

Response: We thank Reviewer #1 for the positive comments. As discussed in the methods part of the revision this time, the isolation of mRNA was performed by using the Dynabeads™ mRNA Purification Kit (ThermoFisher, Catalog number: 61006) according to the manufacturer's protocol. The mRNAs we enriched account for around 4% of total RNAs, suggesting the purity of isolated mRNAs is good. The ratios of A260/A280 for our RNA samples are around 2. In addition, this mRNA Purification Kit was used by many scientists (PMID: 33220174, PubMed ID: 12640466; PubMed ID: 19620212; PubMed ID: 19136943). We also analyzed total RNAs and the enriched mRNAs by the Agilent TapeStation system as shown below, the 18S and 28S peaks were lost in the mRNAs, indicating the depletion of rRNA.

Total RNA

mRNA